

# Measurement report:Characteristics of airborne black carbon-containing particles during the 2021 summer COVID-19 lockdown in Yangzhou, China

Yuan Dai[1,2,3], Junfeng Wang[1,2], Houjun Wang[3], Shijie Cui[1,2], Yunjiang Zhang[1,2],

Haiwei Li[1,2], Yun Wu[1,2], Ming Wang[1,2], Eleonora Aruffo[5], Xinlei Ge[1,2,4*]

[1]Jiangsu Key Laboratory of Atmospheric Environment Monitoring and Pollution Control, Collaborative Innovation Center of Atmospheric Environment and Equipment Technology, School of Environmental Science and Engineering, Nanjing University of Information Science and Technology, Nanjing 210044, China

[2]International Joint Laboratory on Climate and Environment Change (ILCEC), Nanjing University of Information Science and Technology, 210044 Nanjing, China

[3]Yangzhou Environmental Monitoring Center, Yangzhou 225009, China

[4]School of Environment and Energy Engineering, Anhui Jianzhu University, Hefei 230601, China

[5]Department of Advanced Technologies in Medicine & Dentistry, University "G. d'Annunzio" of Chieti-Pescara; Center for Advanced Studies and Technology-CAST, Chieti 66100, Italy

**Correspondence:** Xinlei Ge (caxinra@163.com)





**Abstract**

Black carbon-containing (BCc) particles are pervasive in ambient atmosphere. The unexpected outbreak of the COVID-19 pandemic in 2021 summer prompted a localized and prolonged lockdown in Yangzhou City, situated in the Yangtze River Delta (YRD) region, China, which provides a unique opportunity to gain insights into the relationship between emission sources and BCc. Satellite and ground-level measurements both demonstrated that strict emission controls effectively reduced local gaseous pollutants. Meanwhile, single particle aerosol mass spectrometer (SPA-MS) analysis showed that the number fraction of freshly emitted BCc particles decreased to 28% during the lockdown (LD), with that from vehicle emissions experiencing the most substantial reduction. However, the uncontrolled reductions of nitrogen oxides ($NO_x$) and volatile organic compounds (VOCs) likely contributed to increased ozone ($O_3$) concentrations, increased the oxidizing capacity, which may in turn enhanced secondary $PM_{2.5}$ formation and compensated the primary $PM_{2.5}$ reduction. As a result, we did observe a slight increase of $PM_{2.5}$ concentration (21.2 μg m$^{-3}$) during the LD period compared to the period before the lockdown (BLD), and the increase of more aged BCc particles. Reactive trace gases (e.g., $NO_x$, $SO_2$, and VOCs) could form thick coatings on pre-existing particles likely via enhanced heterogeneous hydrolysis under high RH as well, resulting in significant BCc particle growth (~600 nm) during LD period. Furthermore, BCc source apportionment reveals that BCc particles were primarily of local origin (78%) in Yangzhou during normal summertime. However, coal combustion (23%) and vehicle emissions (21%) were prominent non-local pollution sources, with the air mass originating from the southeast, along with biomass burning emissions (19%) from the northeast, contributing significantly. Our research highlights that short-term, strict local emission controls may not effectively reduce PM pollution, due to the non-linear responses of $PM_{2.5}$ to its precursors , further effective $PM_{2.5}$ reduction requires a comprehensive and extensive approach with a regionally coordinated and balanced control strategy through joint regulation.

## 1. Introduction

China has implemented long-term clean air measures to cut down anthropogenic emissions and improve air quality (Ge et al., 2020), resulting in a nationwide reduction of average fine particulate matter ($PM_{2.5}$, aerodynamic diameter ≤ 2.5 μm) level from 50 μg m$^{-3}$ in 2015 to 30 μg m$^{-3}$ in 2020 (Zhou et al., 2022). However, this $PM_{2.5}$ concentration remains significantly higher than the new World Health Organization (WHO) guideline value of 5 μg m$^{-3}$ (*WHO Global Air Quality Guidelines*, 2021). Black carbon (BC) is a ubiquitous component of $PM_{2.5}$ that can mix with various species, and the number fraction of BC-containing particles (BCc) can be higher than 50% of $PM_{2.5}$ in China (Sun et al., 2022; Xie et al., 2020; Chen et al., 2020). Additionally, the atmospheric aging of BCc involves complex chemical and physical processes, influencing their mixing state, morphology, hygroscopic and optical properties, etc.,



ultimately impacting their climatic and health effects (Bond et al., 2013; Ramanathan
et al., 2008). Reducing the mass loading of BCc is therefore essential to comply with
the new WHO PM$_{2.5}$ guideline. Moreover, the insufficient understanding of complex
emission sources (e.g., fossil fuel and biomass burning), aging processes (e.g.,
coagulation, condensation, and cloud processing), and physical properties (e.g., mixing
state and coating composition) of BCc, hampering the effectiveness of air quality
remediation (Cappa et al., 2019; Kahnert, 2010; Sun et al., 2021).
Yangzhou is located in the central region of Yangtze River Delta (YRD), at the junction
of the Yangtze River and, the Beijing-Hangzhou Grand Canal, which serves as a
prominent economic city, industrial-intensive area, and highly active inland shipping
node in East China. Due to the complex emissions and feedback with the East Asian
monsoons, this region is susceptible to anthropogenic aerosols, especially BCc particles
originating from chemical, steelmaking, coal-fired, petrochemical enterprises, and
transportation, etc. Extensive studies have investigated the responses of atmospheric
pollutants to emission changes during the COVID-19 lockdown measures in the YRD
(Chen et al., 2021; Li et al., 2020; Qin et al., 2021; K. Zhang et al., 2022). However,
the key chemical and physical processes specifically responsible for the BCc particles
in this region are still not fully understood. During the summer of 2021, Yangzhou
experienced a resurgence of COVID-19 with over 500 confirmed cases. In response,
stringent public health measures were imposed from July 28[th] to September 10[th],
including the closure of public transport, and suspension of non-essential industrial
plants, restaurants, shopping malls, and entertainment clubs. People were also
mandated to quarantine at home. Unlike the nationwide COVID-19 lockdown in China
during the cold season of 2020 (Le et al., 2020; Sulaymon et al., 2021b), the summer
lockdown in Yangzhou was more localized but protracted, significantly altering local
anthropogenic emissions while neighboring cities maintained regular operations. This
scenario provides a unique opportunity to explore and compare the diverse mixing
states and aging process of BCc particles in different anthropogenic emission-intensive,
investigate the regional transportation of air pollutants in the YRD, enhance our
knowledge about the formation of BC-associated secondary components (Lei et al.,
2021; Zhang et al., 2020) and understand emissions-meteorology interactions (Jiang et
al., 2021; Le et al., 2020) in the YRD.
Studies on the effects of large-scale and short-term stringent emission control events on
air quality in China have been widely deployed, e.g., the 2008 Beijing Olympic Games
(Wang et al., 2010; Zhou et al., 2010), the 2015 Asia-Pacific Economic Cooperation
(APEC) (Zhu et al., 2015), the 2014 Nanjing Youth Olympic Games (Wang et al., 2022)
and the national COVID-19 lockdown in 2020 winter (Huang et al., 2021; Le et al.,
2020; Li et al., 2020; Wang et al., 2020). Previous studies extensively investigated air
pollutant variations during the COVID-19 lockdown in the winter of 2020 across
different regions of the world. Stringent restrictions on industrial and vehicular



activities have resulted in significant reductions in gaseous pollutants and particulate
matter, not only in megacities (Chen et al., 2020; Jeong et al., 2022; Sun et al., 2020)
but also in middle-sized cities (Clemente et al., 2022; Wang et al., 2021; Xu et al., 2020)
and rural areas (Cui et al., 2021, 2020; Jain et al., 2021). Compared to the decreasing
trends observed in most cities worldwide, the level of $PM_{2.5}$ in Shanghai (Chang et al.,
2020), Hohhot (Zhou et al., 2022), and the Northeast of China Plain (Nie et al., 2021)
increased unexpectedly. These observations reveal the complex aerosol chemistry of
$PM_{2.5}$ comprising primary and secondary components. The reduction of primary
pollutants during lockdown resulted in a shift towards a higher proportion of secondary
aerosols, including inorganic and organic species, exhibiting a non-linear response to
emission changes (Zhang et al., 2021). Furthermore, some studies have suggested that
the increase in secondary aerosols during lockdown is due to the enhanced atmospheric
oxidative capacity resulting from the rise in ozone levels (Wang et al., 2021),
unfavorable meteorological conditions (Chien et al., 2022; Sulaymon et al., 2021a),
changes of local and regional emission sources (Feng et al., 2022), etc. However, most
previous studies focused on lockdown events during the cold seasons, studies on
summer lockdown events in China were very limited.
To better understand the chemical compositions and aging characteristics of airborne
BCc particles in the YRD, we conducted ground measurements, spaceborne
observations, and mass spectrometric analysis during the COVID-19 2021 summer
lockdown in Yangzhou. Besides, We employed potential source contribution function
(PSCF) analysis and a novel approach for distinguishing local sources to study the air
pollution regional transport in the YRD. This study investigated the impact of small-
scale and short-term stringent emission controls on local ambient aerosol and the
mixing state of BCc particles, providing valuable insights for future air pollution control
measures.

## 2. Methods

### 2.1 Sampling site and instruments

The in-situ online measurements were conducted at a rooftop laboratory 20 m above
ground located in a national air quality monitoring station, Yangzhou Environmental
Monitoring Center (32.41ºN, 119.40ºE), Yangzhou, China (Figure 1). This sampling
site is a typical urban site surrounded by residential areas, arterial roads, parks,
restaurants, and shopping centers. In this study, the measurement period was divided
into three phases: the before-lockdown period (BLD: 30 June to 27 July 2021), the
lockdown period (LD: 28 July to 9 September 2021), and the after-lockdown period
(ALD: 10 September to 7 October 2021) (Figure 2).
A single-particle aerosol mass spectrometer (SPA-MS, Hexin Analytical Instrument Co.,
Ltd., China) was deployed during the field campaign to obtain chemical composition,



size distribution, and mixing state of individual $PM_{2.5}$ particles. A $PM_{2.5}$ cyclone (Model
URG-2000-30ED) and a Nafion dryer are equipped in front of the sampling inlet.
Individual particles are introduced into the SPA-MS through a critical orifice with a
flow rate of 3 L min$^{-1}$. The vacuum aerodynamic diameters ($D_{va}$) are determined using
the velocities derived from two continuous laser beams (diode Nd: YAG, 532 nm)
spaced 6 cm apart. Subsequently, these particles are desorbed and ionized by a
downstream pulsed laser (266 nm), and ion fragments are generated and measured by
a Z-shaped bipolar time-of-flight mass spectrometer. A more detailed description of
SPA-MS can be found in previous studies (Li et al., 2011, Zhang et al., 2022).

$PM_{2.5}$ mass concentration was measured by a particulate matter monitor (XHPM2000E,
Xianhe, China). Nitrogen oxides ($NO_x$ = NO + $NO_2$), $SO_2$, and ozone ($O_3$)
concentrations were detected with a set of Thermo Fisher Scientific instruments
(Models 42i, 43i, and 49i). The concentrations of 103 volatile organic compounds
(VOCs) in ambient air, comprising 57 ozone precursors (PAMS), 12 aldehydes and
ketones, and 34 toxic organics (TO15), were continuously monitored at hourly intervals
using an online device (TH-300B, Tianhong, China). Meteorological parameters,
including ambient temperature (T), relative humidity (RH), precipitation (PCP), wind
direction (WD), and wind speed (WS) were observed synchronously using an automatic
weather instrument (WXT530, Vaisala, Finland). All online data presented in this paper
were hourly averaged at local time (Beijing time, UTC+8).



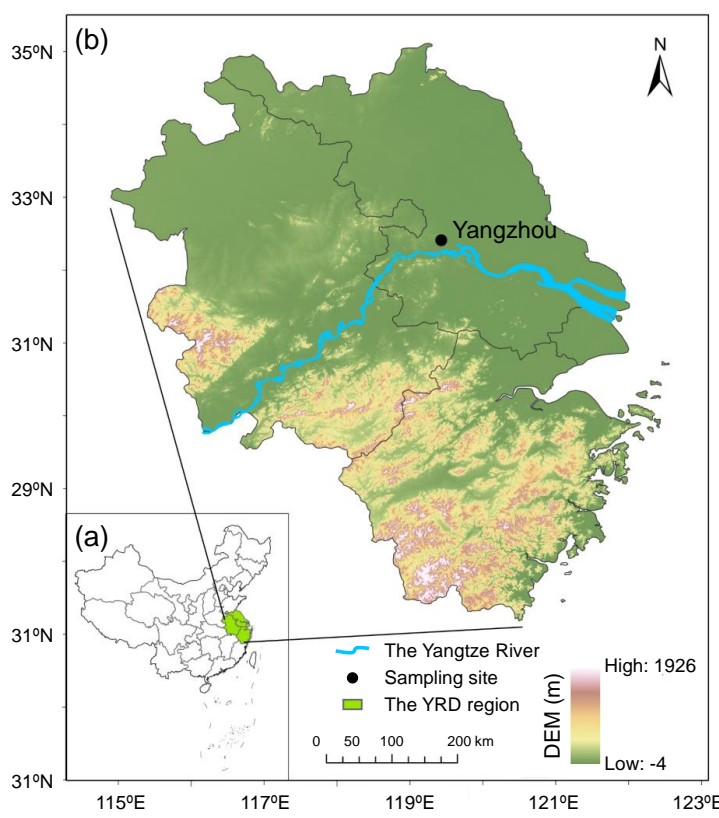

**Figure 1.** Location of **(a)** the Yangtze River Delta (YRD) in China and **(b)** the sampling
site in Yangzhou (Maps were generated by using Arcgis Pro).

## 2.2 Data analysis

### 2.2.1 Satellite Product

Remote sensing of nitrogen dioxide ($NO_2$) and sulfur dioxide ($SO_2$) using satellite has
become a crucial tool for studying air pollution on a large spatial scale. In this study,
we utilized the Level 3 Near Real-Time Product of $NO_2$ (NRTI/L3 NO2) obtained from
the TROPOspheric Monitoring Instrument (TROPOMI) with a spatial resolution of
$3.5 \times 7$ km$^2$ to analyze the distribution of total vertical column of $NO_2$ (Cooper et al.,
2022). To avoid the obvious noises present in the NRTI/L3 $SO_2$ data over clean regions,
we employed the SO2SMASS band from the Modern-Era Retrospective Analysis for
Research and Applications, version 2 (MERRA-2 SO2SMASS) with a spatial
resolution of $69 \times 55$ km$^2$ to represent the distribution of $SO_2$ surface mass concentration
(Ukhov et al., 2020). We calculated and plotted the averaged 2-dimensional data of
NRTI/L3 $NO_2$ and SO2SMASS during BLD and LD over the region of interest
($17.93 \sim 54.74$ ºN, $71.21 \sim 142.23$ ºE) using Google Earth Engine (Gorelick et al., 2017).
The integration of MERRA-2 and TROPOMI measurements has provided a more
comprehensive understanding of the sources and distributions of $NO_2$ and $SO_2$



facilitating the evaluation of the impact of human activities on air quality.

**2.2.2 Geographic Source Analysis**

The potential source contribution function (PSCF) analysis, based on the Hybrid
Single-Particle Lagrangian Integrated Trajectory (HYSPLIT) model, can be employed
to identify regional sources of air pollutants. Before conducting the PSCF analysis, 36
hours of air mass backward trajectories with one-hour resolution at 500 m above ground
level were calculated using the wind data from the Global Data Assimilation System
(GDAS) provided by the National Oceanic and Atmospheric Administration (NOAA)
(Wang et al., 2009). An open-source software MeteoInfo (Wang  et al., 2014) was
utilized for the PSCF analysis. The whole study area (110.1~133.4 ºE and 21.3~39.9
ºN) covered by the trajectories was divided into thousands of cells with a spatial
resolution of 0.1° × 0.1°. The PSCF was simulated according to the following equation:

$$PSCF_{ij} = \frac{m_{ij}}{n_{ij}} \tag{1}$$

where $PSCF_{ij}$ is the conditional probability that the grid cell $(i, j)$ was a source of the
species found in high concentration (Hopke et al., 1993); $n_{ij}$ is the number of all
trajectories passing through this grid cell, and $m_{ij}$ is the number of trajectories. In this
study, the pollution criterion values for different BCc particle types were set as the 75[th]
percentile of hourly average number fractions, respectively. To further improve the
accuracy of the PSCF analysis and minimize analytical uncertainties, the Weighted
PSCF (WPSCF) functions as shown in Equation (2~3) were applied (Polissar et al.,
1999). The weight $(W_{ij})$ for each grid cell was determined based on the number of
trajectory endpoints $(n_{ij})$ as follows:

$$WPSCF_{ij} = W_{ij} \times PSCF_{ij} \tag{2}$$

$$W_{ij} = \begin{cases} 1.00 & n_{ij} > 3n_{ave} \\ 0.70 & 1.5n_{ave} < n_{ij} \leq 3n_{ave} \\ 0.40 & n_{ave} < n_{ij} \leq 1.5n_{ave} \\ 0.17 & n_{ij} \leq n_{ave} \end{cases} \tag{3}$$

Here, $n_{ave}$ is the average number of trajectory endpoints of each grid.

**2.2.3 SPA-MS Data Analysis**

In total, 1649574 particles were analyzed during the entire observation period. The size
and chemical composition of single particles were analyzed using the Computational
Continuation Core (COCO V1.4) toolkit in MATLAB 2022 (The MathWorks, Inc.).
Our focus was on BCc particles, which were identified based on the relative peak area
(RPA) of carbon ion clusters ($C_n^{\pm}$, n = 1, 2, 3, …), with a threshold of 0.05 (Zhang et al.,
2021). An adaptive resonance theory-based neural network algorithm (ART-2a) was
applied to classify the measured individual particles based on the presence and intensity
of ion peaks, with a vigilance factor of 0.75, a learning rate of 0.05, and 20 iterations
(Song et al., 1999).





## 3. Results and discussion

### 3.1 Overview of field observations

Figure 2 presents the temporal variations of meteorological parameters, $PM_{2.5}$, $NO_x$, and $SO_2$ concentrations during the entire observation. During the BLD stage, the mean temperature (T) was 28.2±2.6 °C, with an average relative humidity (RH) of 81.4± 11.1%. The prevailing winds originated from the south and southeast, with a mean wind speed (WS) of 3.4±0.9 m s$^{-1}$. Notably, $PM_{2.5}$, $NO_x$, and $SO_2$ were dramatically reduced at the end of the BLD period due to a high precipitation event, and the data collected during the precipitation were excluded from the analysis. In comparison, the LD period saw a decline in temperature to 26.2±2.4 °C, a reduction in WS to 2.3±0.8 m s$^{-1}$, and an increase in RH to 86.6±10.1%. Additionally, Figures S2b~c exhibit uniform distributions of RH and boundary-layer height (BLH) across the YRD during the LD period. The resemblance of meteorological elements with other cities in the YRD (Qian et al., 2022; Wang et al., 2022) and the effective removal of the pollutants accumulated during the BLD stage imply that Yangzhou is mainly affected by upwind transmission during the LD period, providing favorable conditions for investigating the regional transport of BCc particles in the YRD during summer. Subsequently, the temperature declined to 25.2±3.5 °C, the WS increased to 3.2±1.4 m s$^{-1}$, and RH decreased to a lower level of 74.7±15.0% during the ALD period.

Further, surface concentrations of $NO_x$ (18.9 μg m$^{-3}$) and TVOC (55.0 μg m$^{-3}$) were the lowest during the LD period compared to those of the BLD and ALD periods, whereas the surface $O_3$ concentration showed an increase of 12.6 μg m$^{-3}$ (19%) during the LD period compare to the BLD period, which may attribute to the reduction of fresh NO emissions that alleviates $O_3$ titration (Steinfeld, 1998). However, the average concentrations of $PM_{2.5}$ (19.9 vs. 21.2 μg m$^{-3}$) and $SO_2$ (9.4 vs. 9.5 μg m$^{-3}$) were very close between BLD and LD stages (Figure 3). Following the end of lockdown, social activities gradually resume in Yangzhou City, leading to an apparent increase in all observed pollutants during the ALD period. For instance, the relative increases from LD to ALD were 71% for $NO_x$, 22% for $SO_2$, 55% for TVOC, 30% for $O_3$, and 29% for $PM_{2.5}$ (Figure 3), respectively.





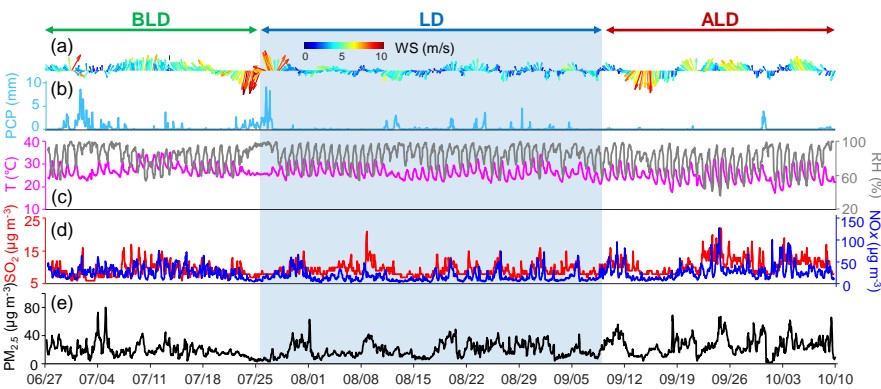

**Figure 2.** Temporal variations of **(a)** wind direction (WD) and wind speed (WS), **(b)** precipitation (PCP), **(c)** temperature (T) and relative humidity (RH), **(d)** concentrations of $NO_x$ and $SO_2$, and **(e)** mass loading of $PM_{2.5}$.

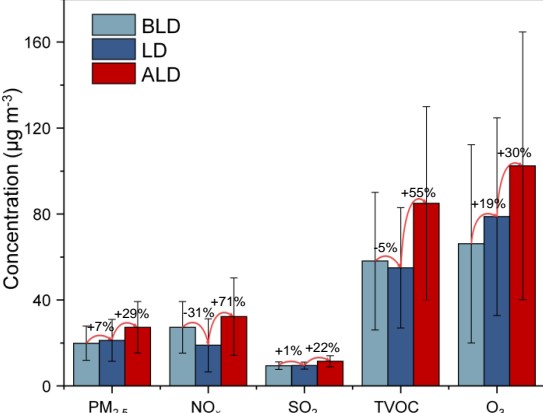

**Figure 3.** Ground-based observations of $PM_{2.5}$, $NO_x$, $SO_2$, $O_3$, and TVOC concentrations in Yangzhou. The figure compares the averages during the BLD (blue-grey), LD (dark-blue), and ALD (crimson) periods. Error bars indicate SDs over different lockdown periods.

In addition to ground measurements, satellite retrieved $PM_{2.5}$, $NO_2$, and $SO_2$ data over the entire region of eastern China were also investigated. Results show that the hotspots of those pollutants were predominantly located over eastern China, e.g., the YRD region, during both the BLD and LD periods (Figure 3). Figure 4 displays the regional fractional changes in mean $PM_{2.5}$, $NO_2$, and $SO_2$ levels from the BLD to LD periods in the YRD, indicating a 29%, 25%, and 23% increase, respectively. In comparison, Yangzhou city experienced lower increases in these air pollutants, with changes of 25%, -23.8%, and 2.9% for $PM_{2.5}$, $NO_2$, and $SO_2$, respectively (remarkable $NO_2$ decrease). Such results highlight the short-term, limited-scale, and human-induced reduction in air pollutants as a result of the lockdown measures in Yangzhou, and demonstrate the





effectiveness of regional stringent emission control in reducing local atmospheric
pollutant concentrations.

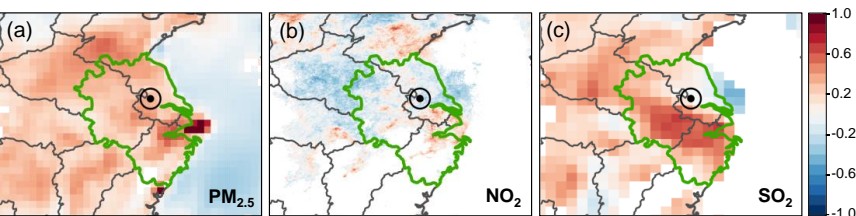

**Figure 4.** The fractional changes (i.e., 100×(LD – BLD)/BLD) of **(a)** PM$_{2.5}$, **(b)** NO$_2$,
and **(c)** SO$_2$ between BLD and LD periods based on spaceborne measurement.
Calculations were only conducted for the regions with PM$_{2.5}$ > 10 µg m$^{-3}$, NO$_2$ > 0.2
Dobson units (DU), and SO$_2$ > 0.2 DU in the BLD period. The circle symbols in the
maps indicate the location of Yangzhou, and the green region represents the YRD (1
DU = 0.4462 mmol m$^{-2}$).

## 3.2 Chemical composition and size distribution of individual BCc particles

Based on the SPA-MS analysis, a total of 1068362 BCc particles were collected over
the whole study period. The BCc particles accounted for 58.8%, 67.7%, and 56.5% of
the total number of measured particles in the BLD, LD, and ALD periods, respectively.
Figure 5 shows the normalized average mass spectra of BCc particles during three
periods. Ion height in each spectrum reflects the number fraction of the detected BCc
particles with the corresponding ion to the total BCc particles, while colors represent
peak area ranges of detected ions. As shown in Figure 5, BCc particles in BLD, LD,
and ALD shown similar mass spectra at *m/z* < 100, with common peaks including
carbon ion clusters (C$_n^{\pm}$, n = 1~7), 27[C$_2$H$_3$]$^+$, 37[C$_3$H]$^+$, 43[C$_2$H$_3$O]$^+$, 51[C$_4$H$_3$]$^+$,
63[C$_5$H$_3$]$^+$, 46[NO$_2$]$^-$, 62[NO$_3$]$^-$, and 97[HSO$_4$]$^-$. However, the abundance of large *m/z*
carbon ions (C$_n^{\pm}$, n > 7) in both BLD and ALD periods was approximately 1.5 times that
in the LD. The result was a clear reflection of less local vehicle emissions during the
LD period (Liu et al., 2019), in line with aromatics, e.g., 119[C$_9$H$_{11}$]$^+$.

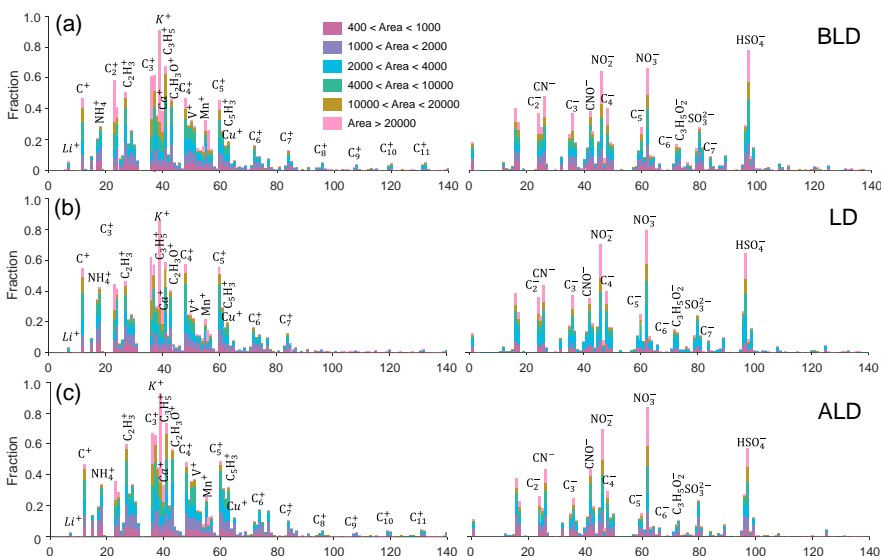

**Figure 5.** The average positive and negative mass spectra of BCc particles **(a)** before the lockdown period (BLD), **(b)** during the lockdown period (LD), and **(c)** after the lockdown period (ALD).

BCc particles were further classified into 12 types based on the differences of chemical features and temporal variations as shown in Table S1. Fresh BC particles (BC-fresh) are those freshly emitted without undergoing significant atmospheric processing (Ding et al., 2021). Five types of BC-fresh particles were identified according to the characteristics ion markers: (i) BC-pure is dominated by carbon clusters ($C_n^{\pm}$) with minor ion signals of secondary inorganic species, such as $46[NO_2]^-$ and $97[HSO_4]^-$ from nitrate and sulfate, respectively (Xie et al., 2020); (ii) BCc particles from biomass burning (BB) are characterized by ion signals at $m/z$ $39[K]^+$, $45[CHO_2]^-$, $59[C_2H_3O_2]^-$, and $73[C_3H_5O_2]^-$, with relative peak area (RPA) more than 0.5 (Silva et al., 1999); (iii) coal combustion BCc particles (CC) typically include small carbon clusters ($C_n^{\pm}$, n = 1~4), metal elements (e.g., $7[Li]^+$, $23[Na]^+$, $27[Al]^+$, $56[Fe]^+$, $63[Cu]^+$ and $206/207/208[Pb]^+$), and organic carbon ($38[C_3H_2]^+$, $43[C_2H_3O]^+$) peaks in the positive mass spectrum, while the strong signals of secondary inorganic species ($46[NO_2]^-$, $43[AlO]^-$, $62[NO_3]^-$, $80[SO_3]^-$, $97[HSO_4]^-$) in the negative ion mode suggest that CC particles were long-distance transported or more processed (Zhang et al., 2022; Zhang et al., 2009); (iv) particles from vehicle emission (VE) are characterized by the presence of ion signals at $m/z$ of $40[Ca]^+$, $51[V]^+$, $55[Mn]^+$, $67[VO]^+$, $46[NO_2]^-$, $62[NO_3]^-$, and $79[PO_3]^-$, as well as high loadings of organic carbon ($41[C_3H_5]^+$, $43[C_2H_3O]^+$) and carbon clusters ($C_n^{\pm}$, n = 1~4) ion peaks (Yang et al., 2017); (v) BCc particles that are internally mixed with more than one type (BB, CC, and VE) are categorized as Mix type (Sun et al., 2022).





Aged BC particles, denote as BC-aged, undergo a series of chemical reactions and
physical transformations. These processes typically lead to changes in their morphology,
hygroscopicity, and optical properties as they are coated with other materials (He et al.,
2015). Six types of BCc particles are classified as BC-aged and are further grouped into
BCOC and BC-SNA, depending on whether they contain mainly organic carbon (OC)
or sulfate/nitrate/ammonium (SNA). First, BCOC types indicate BC-aged particles that
internally mixed with OC. These particles are characterized by the presence of carbon
clusters ($C_n^{\pm}$) and $C_nH_m^+$ ions (n = 1~6, m = 1~3) in positive mass spectra (Xie et al.,
2020). On the other hand, BC-aged particles that do not mix with OC are named BC-
SNA indicating the mix with secondary inorganic species. Additionally, BCOC
particles with negative mass spectra dominated by nitrate ions ($46[NO_2]^-$ and $62[NO_3]^-$)
or sulfate ions ($97[HSO_4]^-$) are referred to as BCOC-N or BCOC-S, respectively;
otherwise, BCOC particles showing similar peak areas of nitrate and sulfate are named
BCOC-SN. Furthermore, The BC-SNA particles are further categorized as BC-N, BC-
S, and BC-SN based on similar principles. Note the remaining particles that cannot be
classified into neither BC-fresh or BC-aged ones are denoted as BC-other. More details
of BCc particle types are shown in Table S1 and Figure S1 in the Supplement.

During the BLD period, the average number fraction of BC-fresh particles was 36%
with sizes mainly concentrated at ~500 nm, while the mode size of BC-aged particles
was ~520 nm (Figure 7). The predominant BCc types during the BLD period were
BCOC-S and BC-S (24% and 12% by number), likely because sulfate was removed
less efficiently than organic matter (OM) and $NO_3$ by heavy precipitation, especially
during the warm seasons (Isokääntä et al., 2022). As shown in Figure 7c and d, the peak
size of BC-SNA was larger than that of BCOC in all periods, indicating that organics
coated BCc generally had a relatively thin coating compared to those coated by
secondary inorganic species, which is consistent with previous studies (Sun et al., 2016;
Wang et al., 2019).

During the transition of BLD period ($PM_{2.5}$: 19.9 μg m$^{-3}$, $O_3$: 66.2 μg m$^{-3}$, $NO_x$: 27.3 μg
m$^{-3}$) to LD period ($PM_{2.5}$: 21.2 μg m$^{-3}$, $O_3$: 78.8 μg m$^{-3}$, $NO_x$: 18.9 μg m$^{-3}$), heavy
precipitation occurred on July 28[th] (the day before lockdown) and scavenged a majority
of the pollutants ($PM_{2.5}$: 4 μg m$^{-3}$, $O_3$: 35 μg m$^{-3}$, $NO_x$: 8 μg m$^{-3}$). After that, the strict
lockdown measures were carried on and the primary emissions were abruptly cut down.
As a result, the number fraction of BC-fresh particles significantly decreased from 37%
to 28% and that of VE-type particles dropped from 12% to 3% (by number). As shown
in Figure 3, with the decrease in $NO_x$, an obvious enhancement of $O_3$ was observed
during the LD period. According to previous studies (Huang et al., 2021; Laughner et
al., 2021), large reduction of $NO_x$ could promote the formation of $O_3$ under a VOC-
limited regime and enhance the oxidation capacity of the local atmosphere, which made
the number fraction of BC-aged particles increased from 64% in the BLD to 72% in the
LD period (Figure 6a), indicating the lockdown measures could accelerate aging of BCc



particles through complicated chemical reactions and/or physical coagulation. We also
found the most abundant type of BCc particles changed from BCOC-S (24% by number)
in the BLD to BC-N (25%) in the LD (Figure 6a). Furthermore, despite the abrupt
reductions of NO$_x$ due to city lockdown, it should be aware that the PM$_{2.5}$ concentration
slightly increased during the LD period, highlighting the non-linear relationship
between primary emissions and PM$_{2.5}$ levels.
During the ALD period (PM$_{2.5}$: 25.9 µg m$^{-3}$, NO$_x$: 27.9 µg m$^{-3}$, TVOC: 76.4 µg m$^{-3}$),
the number fraction of BC-fresh particles increased from 28% to 31%, and the fraction
of VE particles also increased from 3% to 12% (by number) (Figure 6a). Notably, the
size distributions of BC-fresh and BC-aged particles presented relatively small peaks
at 690 nm and 820 nm during the ALD period, in addition to the prominent peaks at
490 nm and 500 nm, which were different from those in the BLD and LD periods. These
small peaks were relatively close to the dominant sizes of BC-fresh and BC-aged
particles during the LD period (Figure 7). This result suggests that a substantial number
of BCc particles with small sizes (around 500 nm) after the lockdown was lifted in
Yangzhou, owing to the sudden enhancement of primary emissions; on the other hand,
particles with large diameters (>690 nm) may have formed due to the participation of
more trace reactive gases (e.g., NO$_x$, SO$_2$, and VOCs) in continuous aging reactions,
resulting in thicker coatings on the surface of pre-existing particles and therefore a more
clear separation of two mode sizes during the ALD period than during the other two
periods. This hypothesis was also supported by the increased number fraction of
BCOC-SN during the ALD period (Figure 6a). Similar findings have been reported in
the North China Plain (NCP) and the YRD during cold seasons, where thicker coatings
on secondary aerosols were also observed under lower RH (<70%) (Zhang et al., 2021).
This might be due to that particles with more organics and nitrate can result in earlier
deliquescence and provide aqueous surfaces that facilitates heterogeneous formation of
secondary species under relatively low RH (Zhang et al., 2021). Among the three
periods, the difference between the mode sizes of BC-aged and BC-fresh particles was
the smallest (10 nm) during the ALD period (Figure 7a and b). This size reduction can
be attributed to the increased BCOC and hydrophobic primary particles after lockdown
(Figure 6). Because the internally mixed BCOC and hydrophobic primary particles may
constrain further growth of secondary BC-SNA particles (Liu et al., 2016; Zhang et al.,
2018), thereby leading to smaller-sized BC-aged particles. Moreover, the differences of
BCc mode sizes between ALD and BLD periods also reveals an interesting fact that the
lockdown effect may not only affect air quality during lockdown, but also can influence
the air quality even after lockdown, as the resumed emissions after lockdown may be
subjected to different chemistry from that before lockdown.
Throughout the entire observation, the changes in the number fraction of BC-SNA
showed consistency with the variations in RH (Figure 6b), indicating that BC tends to
mix with ammonium sulfate and ammonium nitrate under high RH conditions overall.





Meanwhile, the number fraction of BCOC had similar patterns of change as TVOC,
suggesting that high TVOC levels may facilitate the coating of organics on BC cores.
Figure 8 displays the number fraction of BCc species as a function of PM$_{2.5}$. Overall,
as PM$_{2.5}$ levels increased, the number fraction of BC-aged particles also increased,
while the proportion of BC-fresh particles decreased during the BLD and LD periods,
indicating a clear transition from BC-fresh particles to more aged ones. However, the
increase in PM$_{2.5}$ was driven by BCOC-S during the BLD period (Figure 8a), whereas
BC-N played a vital role in the PM$_{2.5}$ increase during the LD period (Figure 8b).
Interestingly, the concentration of NO$_x$, the primary precursor of BC-N, decreased by
31% and 41% during the LD period compared to the BLD and ALD periods,
respectively (Figure 3). Despite the significant decrease, the proportion of BC-N during
the LD period was unexpectedly higher than those during the BLD and ALD periods,
indicating a strong non-linear response of nitrate in BCc particles to NO$_x$, very likely
due to much faster conversion of NO$_x$ to nitrate upon enhanced atmospheric oxidation
capacity; additionally, the high proportion of BC-N during the LD period might be
attributed to regional transport, similar to that in Shanghai during 2020 winter lockdown
(Chang et al., 2020).

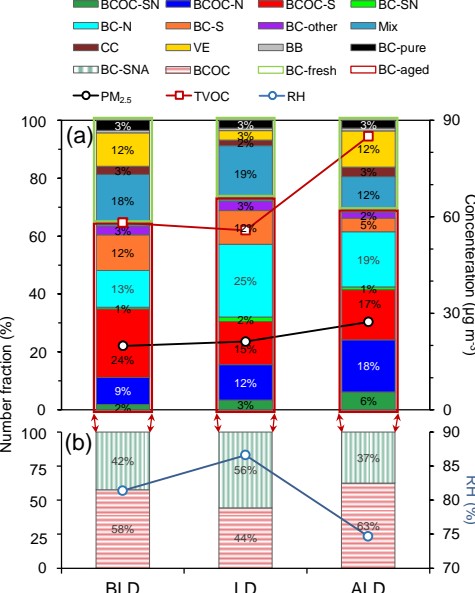

**Figure 6.** Number fractions of BCc particles. **(a)** The number fractions of different BCc
particles along with the concentrations of PM$_{2.5}$ and total volatile organic compounds
(TVOC). **(b)** The number fractions of different types of BC-aged particles along with
relative humidity (RH).





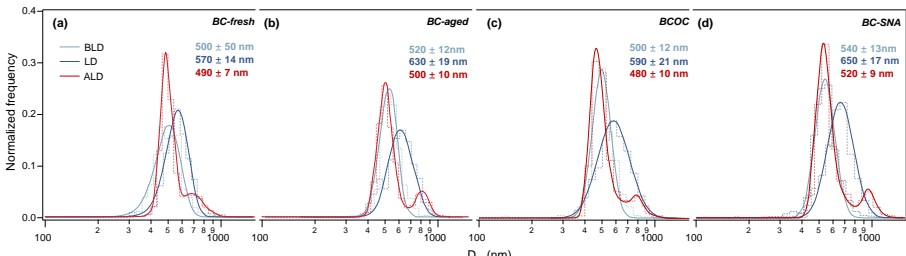

**Figure 7.** Size distribution of different types of BCc particles during different lockdown in Yangzhou. **(a)** BC-fresh particles, **(b)** BC-aged particles, **(c)** BCOC particles, and **(d)** BC-SNA particles. The Log-normal distribution was used to fit the unimodal size distribution, and the Lorentz distribution was used to fit the bimodal size distribution. The corresponding mode sizes (with the standard deviations) are also shown.

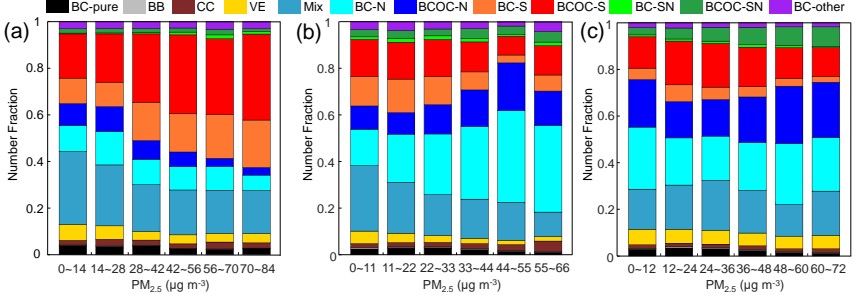

**Figure 8.** Variations of number fractions of BCc particle types with $PM_{2.5}$ mass concentrations during **(a)** the BLD period, **(b)** the LD period, and **(c)** the ALD period.

## 3.3 Chemical aging of BCc Particles

As shown in Figure 5, in the average positive mass spectra of total BCc particles, the peak areas of $C_n^+$, OM, and metals contributed to more than 95% of the total, while nitrate and sulfate peak areas accounted for more than 90% of the negative mass spectral signal. To better elucidate the aging processes of BCc particles during different lockdown periods, we summed the carbon clusters $C_n^\pm$ (n = 1~5, accounting for more than 99% of $C_n$) peak areas to represent BC, and the total peak area of sulfate, nitrate, and ammonium (SNA) to represent the second inorganic components coated on BC. Additionally, we defined the sum of positive peak areas, excluding $C_n^+$ and metals, as OC to represent the OM coated on BC. These peak areas encompassed almost all the coating materials, except for metals, of BCc particles. The changes in the mixing state and morphology of BCc particles can provide insights into their aging characteristics, as reported previously (Kandler et al., 2018; Moffet et al., 2013). In this study, we use $OC/C_n$ and $SNA/C_n$ ratios to describe different types of chemical components coated on BC-fresh, and we use the ratio of the mode size of BC-aged ($D_{aged}$) to that of contemporaneous BC-fresh ($D_{fresh}$) to represent the aging degree of BCc particles.



Figure 9 illustrates the diurnal variations of the $OC/C_n$ and $SNA/C_n$ ratios along with
the size distribution of BCc particles during different periods. Throughout the entire
observation, we observed that both $OC/C_n$ and $SNA/C_n$ increased during nighttime and
decreased during daytime. These variations showed the prominent enhancements of
nocturnal OM and SNA, which could be attributed to the accelerated gas-to-particle
partitioning and nocturnal secondary formation of organic/inorganic components under
high relative humidity (RH > 85%) and relatively stagnant air (WS < 3 m s$^{-1}$) (Figure
S4). It is worth noting that from the BLD period to the LD and ALD periods, the
intensity of diurnal variations of $OC/C_n$ and $SNA/C_n$ increased obviously. This
discrepancy can attribute to several reasons. During the BLD period, the frequent
precipitations effectively scavenged the particles (Isokääntä et al., 2022). In contrast,
stronger solar radiation and higher ozone level during the LD period promoted
photochemical formations of OC and SNA; After lockdown, more precursors due to
increased local emissions may lead to more production of secondary components than
that during the BLD period as explained earlier. These results indicate that the aging
process and mixing state of BCc particles depend strongly on weather conditions and
anthropogenic emission structures in urban cities.
As shown in Figure 9, BCc particles with ~400 nm $D_{va}$ exhibited significant diurnal
fluctuations in the $OC/C_n$ and $SNA/C_n$ ratios, during the LD period. Moreover, there
was a noticeable increase in the proportion of BC-SNA particles during nighttime when
RH was relatively high. These observations suggest that nighttime heterogeneous
hydrolysis may be considered as a key mechanism responsible for the formation of
BCOC and BC-SNA particles. According to Jacobson (2002), coagulation can be
significant between particles with sizes <100 nm and >1 μm but insignificant for
particles of >300 nm, when the total particle number concentration is higher than
$10^4$ cm$^{-3}$. During the LD period, the $OC/C_n$ and $SNA/C_n$ ratios of BCc particles with
~400 nm $D_{va}$ exhibited pronounced diurnal variations (Figure 9) and the number
fraction of BC-SNA increased obviously. Despite the difference between $D_{va}$ and
physical diameter, such results imply that chemical reactions should be considered as
the major pathway for BCOC and BC-SNA particles of ~400 nm $D_{va}$, while the large-
sized BC-aged particles (>1 μm) may be partially from physical coagulation. Moreover,
the larger peak $D_{va}$ (~600 nm) and higher $D_{aged}/D_{fresh}$ ratios (1.11) were observed
compared to those of the BLD (~510 nm, 1.03) and the ALD periods (~500 nm, 1.02)
(Figure 7). Since RH was significantly higher during LD period (average RH of 86.6%)
than the BLD period (average RH of 81.4%) and ALD period (average RH of 74.7%),
this result again supports that aqueous or heterogeneous reactions might play a more
important role to facilitate the chemical conversion of trace reactive gases (e.g., $SO_2$,
$NO_x$, and VOCs) and then formed a thicker coating on the surfaces of BC cores, leading
to evident growth in the size of BCc particles. In addition, this aqueous or
heterogeneous process during the LD period likely converted partially coated particles
to fully thickly coated BCc particles as well (Figure 11).



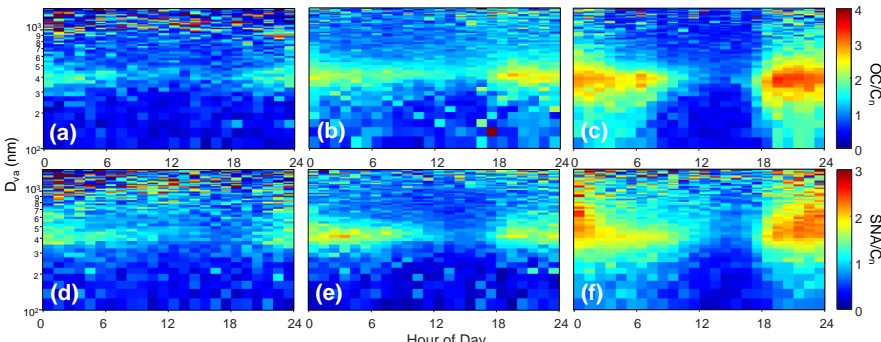


**Figure 9.** Diurnal variations of the ratios of OC/$C_n$ and SNA/$C_n$ with a size distribution
of BCc particles during **(a, d)** the BLD period, **(b, e)** the LD period, and **(c, f)** the ALD
period.

## 3.4 Source of BCc particles during lockdown

In addition to local emissions, regional transport plays a significant role in influencing
the pollutant levels. Due to the emergent lockdown in Yangzhou, local emissions were
strictly limited, while surrounding cities were still as usual, it is therefore interesting to
investigate source areas of BCc sources under such scenario. Besides, the air pollutants
were significantly influenced by regional transport, presenting an ideal opportunity to
investigate the transmission and source characteristics of BCc particles in the YRD
during summer. Here, the PSCF method was used to qualitatively simulate the source
probability distributions of the specific BCc particle types (BC-fresh, BC-aged, BCOC,
and BC-SNA) during the LD period. The results of the potential source regions and
clustering analysis are presented in Figure 10.

As shown in Figure 10, the hotspots of potential sources for the four particle types
exhibited strong agreements with each other and primarily concentrated in the southeast
of Yangzhou, especially along the coast of the Yangtze River, with the WPSCF greater
than 0.6. These hotspot areas also encompassed chemical enterprises, power plants,
petrochemical industrial parks, and the Yangtze River Ship Channel in the YRD.
Moreover, BCc particles and gaseous emissions from the YRD city cluster, heavy
industries, and ship diesel engines, can easily impact the air quality of surrounding
downwind regions. This evidence suggests that the region of southeast Yangzhou and
lower reaches of Yangtze River are major source areas for the regionally transported
BCc particles in Yangzhou during lockdown.



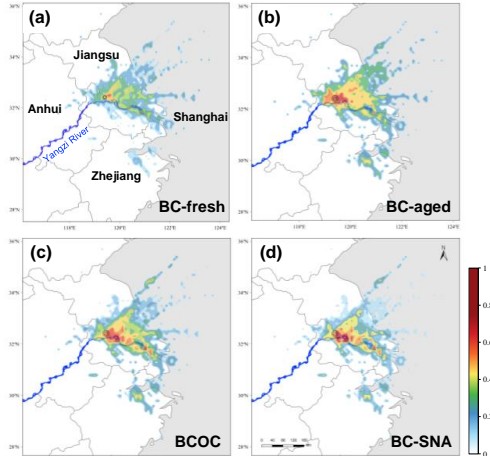

**Figure 10.** The potential sources areas of different BCc particles during the LD period. **(a)** BC-fresh. **(b)** BC-aged. **(c)** BCOC. **(d)** BC-SNA.

According to Luo et al. (2023), regional transport of pollutants can occur near surface from upwind areas when the wind speed (WS) exceeds 2 m s$^{-1}$. Figure S4b shows that the mean daytime WS was 3 m s$^{-1}$, indicating that both BC-fresh and BC-aged particles, along with trace gases (e.g., SO$_2$, NO$_x$, and VOCs), originating from the hotspot areas, could be transported effectively to Yangzhou. Additionally, the average size of BCc particles remained around 0.6 μm at daytime (Figure S4c), suggesting that BCc particles could undergo continual aging reactions under relative lower RH, but producing relatively thinly coated BCc particles with smaller sizes than those at nighttime (average size of 0.65 μm) (Figure 11).The mean nocturnal WS decreased to 2 m s$^{-1}$, indicating that the regional atmosphere becomes stagnant (Figures S4a, b). As mentioned earlier and underscored here again, this stagnant and humid atmospheric condition may promote aqueous or heterogeneous reactions, likely further leading to the production of more thickly coated BCc particles than daytime ones (Figure 11).



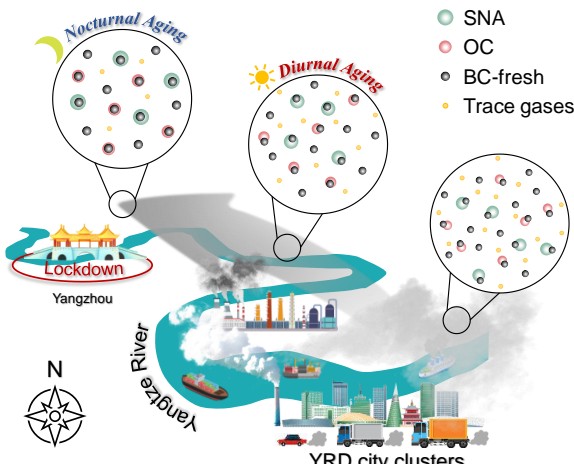

**Figure 11.** A schematic diagram of the transportation of air pollutants and ageing
process from the YRD city cluster to Yangzhou during the 2021 summer COVID-19
lockdown.

## 3.5 Local and non-local source analysis

Since there was a heavy precipitation on July 28[th] (the day before lockdown) which
removed most atmospheric pollutants, the air pollutants might be influenced mostly by
regional transport as local emissions were significantly cut down during the LD period.
As a comparison, the pollution during the ALD period might be caused by both local
emissions and regional transport. Additionally, the meteorological conditions were
relatively stable during the LD and ALD periods (Figure 2a~c); the trajectories of these
two periods were both categorized into similar 3 clusters, indicating stable regional
transport of the pollutants from the southeast, southwest, and northeast (Figure 12). The
lockdown event with favorable meteorological conditions provided a valuable
opportunity to investigate emissions-meteorology interactions in YRD during summer.
Here, we propose a method to roughly estimate the local and non-local proportions of
for each type of BCc particles in Yangzhou during the ALD period (representing the
usual emission condition)
$$[PM]_{i,j}^{non\text{-}local} = [PM]_{i,j}^{LD} \times t_i^{ALD}/t_i^{LD}; \qquad (4)$$
$$[PM]_{i,j}^{local} = [PM]_{i,j}^{ALD} - [PM]_{i,j}^{non\text{-}local}. \qquad (5)$$
For Equation (4~5), the duration of the $i$[th] cluster in different periods is represented by
$t_i^{LD}$ and $t_i^{ALD}$. The sum of the hourly number density of the $j$[th] type of particulate matter
in the $i$[th] cluster during the LD period is denoted as $[PM]_{i,j}^{LD}$. Similarly, $[PM]_{i,j}^{non\text{-}local}$
and $[PM]_{i,j}^{local}$ indicate the summed hourly number density of specific types of BCc
particles from non-local and local sources in the $i$[th] cluster during the ALD period,
respectively.

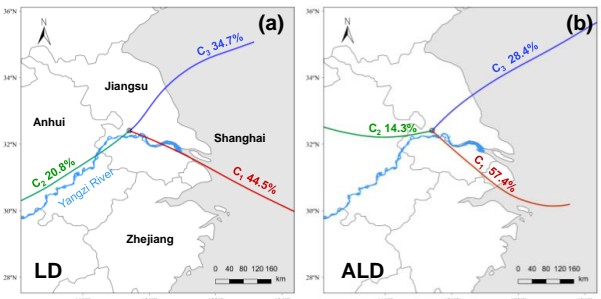

**Figure 12.** Back-trajectory analysis during **(a)** LD and **(b)** ALD period. The corresponding percentages of total trajectories for Cluster1 ($C_1$, red), Cluster2 ($C_2$, green) and Cluster3 ($C_3$, blue) are also shown.

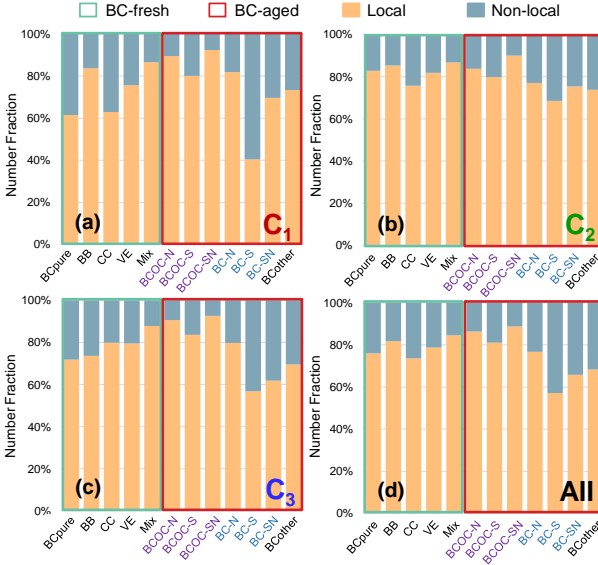

**Figure 13.** Number fraction of local and non-local in different types of particles in **(a)** Cluster1 ($C_1$), **(b)** Cluster2 ($C_2$), **(c)** Cluster3 ($C_3$), and **(d)** all clusters during the ALD period. The purple labels represent BCOC particles and the blue labels represent BC-SNA particles.

By using this method, analysis results of the sources of all types of BCc particles from different clusters are presented in Figure 13a~c. Cluster1 ($C_1$), originating from the economically developed southeast region, accounted for 57.4% of total trajectories during the ALD period. $C_1$ exhibited the highest proportions of non-local fresh BCc (BC-pure, 38.6%) and coal combustion BCc (CC 37.1%), along with vehicular ones (VE 24.3%), underscoring the substantial impact of heavy industry and transportation from the southeastern YRD during summer (Figure 13b). Notably, non-local contribution of BC-S was dominant in $C_1$, indicating significant regionally transported sulfate. Cluster2 ($C_2$), originating from the southwest region covering Nanjing and



eastern Anhui province, accounted for 14.3% of all trajectories (Figure 12b). The non-
local proportions for all particle types were around 20%, indicating that local emission
was dominant (Figure 13b). Cluster 3 (C$_3$), originating from the East China Sea and
passing through the vast cultivated area in northeastern Jiangsu Province, contributed
28.4% of the total trajectories (Figure 12b). A relatively high proportion of BCc
particles generated by non-local biomass-burning emissions (BB, 26.4%) was observed
in C$_3$, indicating a correlation between BCc particles from the northeastern YRD and
open-field burning of agricultural residues during summer (Figure 13c).
Regarding the number fraction of local and non-local contributions for the whole
campaign (Figure 13d), the proportion of local BC-N particles (~80%) exceeded that
of BC-S (~60%; note non-local contribution of BC-S even dominated over local
counterpart in C$_1$), suggesting that sulfate-associated BCc particles were more likely
from regional transport than those of nitrate-associated ones. The proportion of non-
local BC-aged particles was relatively higher than the BC-fresh particles naturally, as
BC-aged particles intercepted more secondary species during the regional transport
than freshly emitted BCc particles. Furthermore, the proportion of local BCOC particles
exceeded that of BC-SNA particles, implying a strong relationship between BCOC
particles and local emission, whereas more BC-SNA particles were likely associated
with regional transport. Overall, BCc particles was predominantly local (78%) in
Yangzhou during normal summertime. However, BCc particle from coal combustion
(CC, 26%) and vehicle emission (VE, 21%) transported from the southeast, as well as
biomass burning-related emissions (BB, 19%) from the northeast, were also significant
contributors that should not be ignored (Figure 13d). These findings highlight the
importance of considering both local and regional sources, as well as understanding the
transport characteristics of different types of BCc particles for air quality management.

## 4. Conclusions and implications

During the summer of 2021, the COVID-19 lockdown imposed in Yangzhou resulted
in a significant decrease in anthropogenic emissions from traffic and manufacturing
sectors. To examine the effects of this lockdown, we utilized spaceborne observations,
ground-based measurements, and particularly SPA-MS analysis to explore the
variations, aging characteristics, and sources of BCc particles in the YRD. We showed
that the strict emission controls effectively reduced local gaseous pollutants. However,
the decline in NO$_x$ (-30.1%) and TVOC (-5.3%) levels might on the other hand result
in increased ozone (+19.0%), leading to a rise in BC-aged particles and a slight
elevation in PM$_{2.5}$ levels during the lockdown. Our results revealed a strong non-linear
response of PM$_{2.5}$ and O$_3$ to the gaseous precursors.
The SPA-MS analysis results further demonstrate significant enhancement of OM and
SNA coating species on BC-fresh particles, owing to gas-to-particle partitioning and
nocturnal multiphase chemistry. Consequently, we observed a higher fraction of BC-



aged particles (73%) during the lockdown due to enhanced oxidizing capacity and high relative humidity (RH > 85%). The BC-fresh particles tended to mix with SNA under high RH conditions, while high TVOC levels were accompanied by BCOC formation. However, BCOC particles generally exhibited smaller sizes compared to BC-SNA particles. Moreover, we postulate that aqueous or heterogeneous reactions might be important to generate BCOC and BC-SNA particles, especially ones with ~400 nm $D_{va}$, while coagulation might play a more prominent role in larger BC-aged particles. The aging process during the LD period promoted the conversion of partly coated particles to totally coated ones, with larger diameters (~600 nm) and thicker coatings.

Furthermore, our study highlights that local emissions were the main source of BCc particles in Yangzhou during normal summertime. However, regional transported coal combustion (23%) and vehicle emissions (21%) from southeast, as well as biomass burning emissions (19%) from the northeast, were also significant. Meteorological conditions, including wind patterns and relative humidity, also influenced the regional transport of BCc particles in the YRD.

It should be noted that the observed average $PM_{2.5}$ concentration during the lockdown in Yangzhou was 21.2 μg m$^{-3}$, which still significantly exceeds the WHO's air quality guideline of 5 μg m$^{-3}$. Our research highlights that reduction of local primary emissions from traffic and manufacturing sectors alone has limited effect in air quality remediation. Complex chemistry, regional transport and meteorological factors need to be considered cooperatively. Therefore, we suggest a more comprehensive regulation of precursor gases from multiple sectors, a wide-ranging joint regulation approach as well as proper consideration of the chemistry, so as to develop an effective strategy for air quality improvement.

**Data availability.** Data described in this manuscript can be accessed at repository under: https://doi.org/10.6084/m9.figshare.24427795 (Dai, 2023)

**Author contributions.** XG and YD designed the research. YD, HW and SC conducted the field measurements. YD, HW, JW and SC analyzed the data. XG, JW, HL, YW, YZ, and EA reviewed the paper and provided useful suggestions. YD and XG wrote the first draft of paper. All people involve in discussion of the results.

**Supplement.** The supplement related to this article is available online at: XXX.

**Competing interests.** The contact author has declared that neither they nor their co-authors have any competing interests.

**Financial support.** This research has been supported by the National Natural Science Foundation of China (grant nos. 42377100 and 42021004).



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
