# Peer review of "Measurement report: Characteristics of airborne black carbon-containing particles during the 2021 summer COVID-19 lockdown in Yangzhou, China"

_EGUsphere, 2023_

## Author Response (AR1)

Responses to reviewers' comments

We thank the reviewers for their detailed, helpful, and overall supportive comments. We have revised the manuscript to account for each comment. Responses to the individual comments are provided below. Reviewer comments are in **bold**. Author responses are in plain text. Modifications to the manuscript are in *italics*. Line numbers in the response correspond to those in the revised manuscript text file.

**Reviewer #1:**

**1. (Figure 3) Do the authors have BC and CO concentration data? Both these chemical species are emitted from incomplete combustion. Their emission ratios depend on types of sources. If the authors could provide these data, it may help supporting the conclusion that the major emission source of BC during the LD period was different.**

We thank the reviewer for insightful comments that help improve the original manuscript. We have incorporated the concentration data of CO into Figure 3 and expanded the original discussion as follows:

[Figure]

Lines 257-261: "*Given that both BC and CO are byproducts of incomplete combustion of carbon-containing fuels (Wang et al., 2015), , and the high correlation between BC*

*and CO (Zhou et al., 2009), it is plausible to infer that the primary emission source of BC during LD differed from that during ALD."*

**2. (Figure 5) The label for x-axis is missing. During the LD period, most of carbon cluster ions were smaller than C7+, while C8+~C11+ ions were abundant during the before/after the LD period. Does the change in the carbon cluster ion sizes tell anything about emission sources?**

We are grateful for the suggestion. We have supplemented the labels for the x-axis in Figure 5 and the text is updated as follows:

Lines 291~295: *"Previous studies have indicated that high-mass carbon ions may be linked to traffic emissions, particularly those from diesel trucks (Xie et al., 2020; Liu et al., 2019), and the observed reduction in such ions during the LD period suggests a decrease in local vehicle emissions. This trend is also consistent with the changes observed in aromatic compounds, e.g. $119[C_9H_{11}]^+$."*

**3. (Line 308) Can these organic ion fragments be produced from POA, SOA, or both of them? Do the authors have any comments?**

Thanks for the comment. We think organic ion fragments can be produced from both POA and SOA. Primary organic aerosols originate directly from emissions sources such as combustion processes (e.g., vehicle exhaust, biomass burning) and can contain a variety of organic compounds. Secondary organic aerosols, on the other hand, are formed in the atmosphere through the oxidation of volatile organic compounds (VOCs) and subsequent condensation onto pre-existing particles. Both POA and SOA can undergo ionization and fragmentation processes, leading to the production of organic ion fragments detected by instruments like mass spectrometers. Therefore, the presence of organic ion fragments in atmospheric aerosols can be indicative of contributions from both primary and secondary sources.

**4. (Line 492) Could condensation also contribute to the process? Or, do the authors have a strong support to demonstrate that the process was dominantly occurring by aqueous or heterogeneous reactions?**

We are grateful for the suggestion. We think condensation can contribute to the process of aerosol formation and growth, particularly in the context of secondary organic aerosol (SOA) formation. However, we also suggest that aqueous or heterogeneous reactions may be the dominant mechanisms driving the observed changes in BCc particles during the LD period.

We do not have direct evident to support this suggestion, there are several lines of clues:

a. **Observations of diurnal fluctuations:** Significant diurnal fluctuations are observed in the OC/Cn and SNA/Cn ratios of BCc particles during the LD period. These fluctuations suggest dynamic chemical processes rather than simple condensation, as condensation alone would not typically exhibit such pronounced diurnal variations.

b. **Increase in BC-SNA particles during nighttime:** There is a noticeable increase in the proportion of BC-SNA particles during nighttime, especially when relative humidity is relatively high. This observation suggests the involvement of heterogeneous hydrolysis, a type of chemical reaction, rather than purely condensation.

c. **Comparison of particle sizes and ratios:** We compared the OC/Cn and SNA/Cn ratios of BCc particles with different diameters and found pronounced diurnal variations in these ratios for BCc particles with a diameter of ~400 nm during the LD period, indicating chemical reactions as the major pathway for particle formation and growth.

d. **Role of relative humidity:** The significantly higher average relative humidity during the LD period compared to the periods before and after suggests favorable conditions for aqueous or heterogeneous reactions. This supports the idea that

chemical conversion of trace reactive gases and the formation of thicker coatings on BCc particles are driven by these processes during the LD period.

**5. Was Yangzhou the only one city which experienced the lockdown in the YRD region during the 3.5 months of the observation period? Could the authors be able to provide some supporting information for the statement? (Line 504~505)**

Thanks for the comment. As far as we know, Yangzhou experienced the most severe epidemic in the Yangtze River Delta (YRD) region during the observation period. Strict city-wide lockdown measures were implemented in Yangzhou, encompassing all downtown communities, from July 28 to September 10, 2021. In contrast, other regions in the YRD, such as Shanghai and Nanjing, implemented lockdown measures in only a limited number of communities during the same period.

**References:**

Liu, D., Joshi, R., Wang, J., Yu, C., Allan, J.D., Coe, H., Flynn, M.J., Xie, C., Lee, J., Squires, F., Kotthaus, S., Grimmond, S., Ge, X., Sun, Y., Fu, P., 2019. Contrasting physical properties of black carbon in urban Beijing between winter and summer. Atmospheric Chemistry and Physics 19, 6749–6769. https://doi.org/10.5194/acp-19-6749-2019

Wang, Q., Liu, S., Zhou, Y., Cao, J., Han, Y., Ni, H., Zhang, N., Huang, R., 2015. Characteristics of Black Carbon Aerosol during the Chinese Lunar Year and Weekdays in Xi'an, China. Atmosphere 6, 195–208. https://doi.org/10.3390/atmos6020195

Xie, C., He, Y., Lei, L., Zhou, W., Liu, J., Wang, Q., Xu, W., Qiu, Y., Zhao, J., Sun, J., Li, L., Li, M., Zhou, Z., Fu, P., Wang, Z., Sun, Y., 2020. Contrasting mixing state of black carbon-containing particles in summer and winter in Beijing. Environmental Pollution 263, 114455. https://doi.org/10.1016/j.envpol.2020.114455

Zhou, X., Gao, J., Wang, T., Wu, W., Wang, W., 2009. Measurement of black carbon aerosols near two Chinese megacities and the implications for improving emission inventories. Atmospheric Environment 43, 3918–3924. https://doi.org/10.1016/j.atmosenv.2009.04.062

**Reviewer #2:**

**1. (Line 29) What kinds of "local gas pollutants"?**

Thanks for your comment. Analysis of satellite and ground-based measurements reveals a reduction in the concentrations of $NO_x$, $SO_2$, and VOCs during the lockdown period. However, the concept of local gas pollutants is imprecise, and the analysis of satellite measurements is not the primary focus of this study. Therefore, we have decided to remove this sentence.

**2. (Line 28-36) These sentences are organized in a little messy. Need to be rewritten more clearly.**

We apologize for the language problems in the original manuscript. We have simplified the text, and the revised version is as follows:

Line 29~33: "*Single particle aerosol mass spectrometer (SPA-MS) analysis revealed a notable decrease in the proportion of freshly emitted BCc during the lockdown period (LD). However, we did observe a concurrent 7% increase in PM$_{2.5}$ concentration during LD, with a higher proportion of aged BCc compared to the period before the lockdown (BLD).* "

**3. (Line 46~50) These highlights do not match the title and the major objects of this paper. The author should provide some guidance for BCc, rather than a general PM$_{2.5}$.**

We agree with the comment and re-wrote the sentence in the revised manuscript as the follows:

Line 38~43: "*Our research highlights that short-term, strict local emission controls may not effectively reduce PM pollution due to the complex generation and transmission characteristics of BCc and the non-linear responses of PM$_{2.5}$ to its*

*precursors. Achieving further effective PM₂.₅ reduction mandates a focus on nuanced control of BCc particles and necessitates a comprehensive and extensive approach with a regionally coordinated and balanced control strategy through joint regulation.*"

**4.  (Line 75-76) References for "Due to the complex emissions and feedback with the East Asian monsoon".**

Thanks for the comment. We have incorporated your suggestion by adding the reference (Ding et al., 2019) in Line 98.

**5.  (Line 122) Any previous results about BCc, as BCc is the topic of your study.**

We appreciate the insightful comment and have incorporated a review of previous studies on BCc into the revised manuscript as the follows:

Line 61~83: " *The atmospheric aging of BCc involves intricate chemical and physical transformations that influence their mixing state, morphology, hygroscopicity, and optical properties, all of which have profound implications for climate and human health (Bond et al., 2013; Ramanathan et al., 2008). For example, freshly emitted BC particles are initially hydrophobic but possess a porous surface structure that facilitates the internal or external mixing with co-emitted primary organic/inorganic and secondary materials that are associated with BC (Cheng et al., 2012; Li et al., 2020). On the other hand, BCc undergoes continually aging processes, including the condensation of low-volatility vapors (Li et al., 2022), coagulation with preexisting aerosols (Kondo et al., 2011), and heterogeneous oxidation with gaseous pollutants (Zhang et al., 2024). This alteration may affect the coating thickness, morphology, size distribution, and hygroscopicity of BCc, thereby impacting their climate forcing as well as atmospheric lifetime (Luo et al., 2022; Taylor et al., 2014). High loading of atmospheric BCc could also depress the development of the planetary boundary layer and exacerbate PM pollution episodes (Huang et al., 2018). BCc characteristics are influenced by various combustion sources and emission conditions, including local*

*industrial burning, vehicle exhausts, residential coal burning, and biomass burning (Li et al., 2020; Sedlacek et al., 2022; Zhang et al., 2018), as well as long-range transport from other regions (Adachi et al., 2014; Zhang et al., 2021). Those diverse conditions complicate the development of parameterizations of BCc properties, the insufficient understanding of complex emission sources, aging processes, and physical properties of BCc, hampering the effectiveness of air quality remediation (Cappa et al., 2019; Kahnert, 2010; Sun et al., 2021). "*

**6. (Line 125) You do not conduct satellite measurements. Change to "combine".**

We agree with the comment and re-wrote the sentence in the revised manuscript as the following:

Line 148: "*Our investigation involved a combination of ground measurements, spaceborne observations, and mass spectrometric analysis conducted during the COVID-19 lockdown in the summer of 2021 in Yangzhou.*"

**7. (Line 48) What is the flow rate and cut size for your cyclone? More detailed information is needed.**

We are grateful for the suggestion. As suggested by the reviewer, we have added more details of the experiment equipment, as outlined below:

Line 153~155: "*A cyclone with 2.5 μm cutpoint (Model URG-2000-30ED) and a Nafion dryer is equipped in front of the sampling inlet. Individual particles are introduced into the SPA-MS through a critical orifice at a flow rate of 3 L min$^{-1}$.*"

**8. (Line 168) Figure 1. Remove (b) and the right corner Chinese map is not very clear. Also the author should add some cities mentioned in the text.**

We are grateful for the comment. We have implemented your suggestion by removing the label (b) and updating Figure 1 with a clearer Chinese map. Additionally, we have

added the names of provinces and cities mentioned in the text to Figure1, as follows:

Line 193~196: "

[Figure]

*Figure 1. Geographical overview of the Yangtze River Delta (YRD) Region in China, depicting the major cities within the YRD and the sampling site located in Yangzhou. The color gradient from green to white indicates varying altitudes across the region (Maps were generated by using Arcgis Pro).*"

**9. (Line 178~182) It is very unclear here. Do you mean that you used the MERRA-2 data to replace the dataset from background SO2 from TROPOMI? Also, please provide the link for your data source and change SO2 mass concentration to SO2 column concentration, as satellite only provides column concentration.**

We apologize for the unclear statement in the original manuscript. We acknowledge that the bands in TROPOMI and MERRA-2 have different spatial resolutions and units, making it challenging to compare remote sensing results from two different sensors. Therefore, we have updated the remote sensing data source to the Copernicus Atmosphere Monitoring Service (CAMS) Global Near-Real-Time dataset for analyzing the distribution of $NO_2$, $SO_2$ and $PM_{2.5}$. Section 2.2.1 has been refined accordingly, as

follows:

Line 177~291: "*In this study, we utilized the Copernicus Atmosphere Monitoring Service (CAMS) Global Near-Real-Time dataset (available at https://developers.google.com/earth-engine/datasets/catalog/ECMWF_CAMS_NRT), acquired from the European Centre for Medium-Range Weather Forecasts (ECMWF), to analyze the distribution of total surface column concentrations of $NO_2$, $SO_2$ and surface $PM_{2.5}$ mass concentration. CAMS offers the capacity to continuously monitor the composition of the Earth's atmosphere at global and regional scales since 2016, with a spatial resolution of 44528 meters (Benedetti et al., 2009; Morcrette et al., 2009). The details of the bands of the dataset used in this study are shown in Table 1. We calculated and plotted the averaged 2-dimensional data of ECMWF/CAMS/NRT $NO_2$, $SO_2$ and $PM_{2.5}$ during BLD and LD over the region of interest (17.93~54.74 ºN, 71.21~142.23 ºE) using Google Earth Engine (Gorelick et al., 2017). The integration of remote sensing measurements has provided a more comprehensive understanding of the sources and distributions of particle matter and gaseous pollutants facilitating the evaluation of the impact of human activities on air quality.*

**Table S2** *Bands of $PM_{2.5}$, $SO_2$ and $NO_2$ in Copernicus Atmosphere Monitoring Service (CAMS) Global Near-Real-Time dataset SP*

| Band Names | Units | Description |
|---|---|---|
| *particulate_matter_d_less_than_25_um_surface* | $kg\ m^{-3}$ | *Surface mass concentration of $PM_{2.5}$* |
| *total_column_nitrogen_dioxide_surface* | $kg\ m^{-2}$ | *Total surface column concentration of $NO_2$* |
| *total_column_sulphur_dioxide_surface* | $kg\ m^{-2}$ | *Total surface column concentration of $SO_2$* |

"

**10. (Line 242) Remove "Further".**

Thanks for the comment. The word "Further" has been removed as requested.

**11. (Line 254) Please provide the meaning about the marked region.**

We are grateful for the suggestion. We have added more details in the caption of Figure 2, as follows:

Line 295~296: "*The grey, blue, and red arrow ranges denote the periods before lockdown (BLD), during lockdown (LD), and after lockdown (ALD).*"

**12. (Line 259) The resolution of Figure 3 is too low.**

We apologize for the unclear figures in the original manuscript. We have refined Figure 3 to ensure high resolution and clarity.

Line 298~303: "

[Figure]

*Figure 3. Ground-based observations of PM$_{2.5}$, NO$_x$, SO$_2$, O$_3$, CO, and TVOC concentrations in Yangzhou. The figure compares the averages during the BLD (blue-grey), LD (dark-blue), and ALD (crimson) periods. Error bars indicate SDs over different lockdown periods.*"

**13. (Line 262) Which satellite produce of PM$_{2.5}$ was used? The related information should be provided in section "2.2.1 Satellite Product".**

We apologize for the missing information in the original manuscript. In order to unify the data source, we have updated the remote sensing data to the Copernicus Atmosphere

Monitoring Service (CAMS) Global Near-Real-Time dataset for analyzing the distribution of NO$_2$, SO$_2$, and PM$_{2.5}$. Section 2.2.1 has been refined accordingly.

**14. (Line 430) Figure 7. It is hard to see the Log-normal distribution.**

We apologize for the unclear figures in the original manuscript. We have revised Figure 7 to enhance clarity and legibility by improving resolution and enlarging the font size Line 489~496: "

[Figure]

*Figure 1.* *Size distribution of different types of BCc particles during different lockdown in Yangzhou.* (*a*) *BC-fresh particles,* (*b*) *BC-aged particles,* (*c*) *BCOC particles, and* (*d*) *BC-SNA particles. The Log-normal distribution was used to fit the unimodal size distribution, and the Lorentz distribution was used to fit the bimodal size distribution. The corresponding mode sizes (with the standard deviations) are also shown.*"

**15. (Line 525) The resolution of Figure 10 is too low.**

We apologize for the unclear figures in the original manuscript. We have refined Figure 10 to ensure high resolution and clarity.
Line 588~590: "

[Figure]

*Figure 2*. *The PSCF maps for different BCc particles during the LD period.* *(a)* *BC-fresh.* *(b)* *BC-aged.* *(c)* *BCOC.* *(d)* *BC-SNA.* "

**16. (Line 541) Figure 11 should be removed and be used as a TOC/Abstract graphic.**

We fully agree with your suggestion. Figure 11 has been removed and will be repurposed as the abstract graphic.

**17. (Line 567) The resolution of Figure 12 is too low.**

We apologize for the unclear figures in the original manuscript. In response to feedback from reviewer #3, we have decided to remove Section 3.5 entirely, along with Figure 12. This decision was made to streamline the manuscript and focus on the most essential findings.

**18. (Line 643-651) Similar to #3, the author should highlight some discussion about BCc, rather than a general PM$_{2.5}$.**

Thank you for your insightful comment. We have revised the conclusion in accordance with your suggestion. The revised sentence reads as follows:

[revised manuscript text omitted]

**Reviewer #3:**

**Major Comments:**

**1. The authors claimed that "the number fraction of freshly emitted BCc particles decreased to 28% during the lockdown (LD), with that from vehicle emissions experiencing the most substantial reduction", and also expressed that "the uncontrolled reductions of nitrogen oxides (NOx) and volatile organic compounds (VOCs) likely contributed to increased ozone (O3) concentrations, increased the oxidizing capacity". My question is what are the emission sources of NOx and VOCs under the scenario of "the vehicle emissions experiencing the most substantial reduction"? Therefore, I think the authors analysis is paradoxical. Therefore, the mechanism of increase of more aged BCs particles is unreasonable.**

Thans for the insightful comment. The emissions of $NO_x$ and VOCs originate from various sources, including vehicles, industrial processes, power plants, and agricultural activities. Despite notable reductions in vehicle emissions, these other sources may still play a significant role in influencing atmospheric composition.

Furthermore, as depicted in **Figure S4**, the levels of $NO_2$ in the major cities of the YRD was significantly higher than that in Yangzhou during the LD period. While $NO_x$ emissions are primarily a local concern, they can also be transported over considerable distances via air currents, as documented in previous studies (Hertel et al., 2012).

Additionally, the elevated ozone levels observed in Yangzhou relative to other major cities in the YRD (**Figure S4** ) underscore the regional variability in atmospheric oxidation processes during the LD period.

Overall, the intricate interplay between diverse emission sources and atmospheric processes presents a valuable opportunity to investigate the characteristics of BCc particles and assess the impacts of lockdown measures on regional air quality. Such analyses are essential for developing effective strategies to mitigate air pollution and protect public health.

[Figure]

**Figure S4.** The ratios of gaseous and particulate levels in the major cities of the YRD compared to those in Yangzhou during the LD period. The black dashed line represents the pollution levels of Yangzhou.

**2. During the observation period, precipitation occurred intermittently (Figure 2b), and the author only mentioned "the data collected during the precipitation were excluded from the analysis" in Section 3.1. This method is obviously not enough to eliminate the impact of precipitation. Additionally, the maximum daily precipitation at the observation site does not exceed 10 mm (Figure 2b), however, the author has repeatedly mentioned heavy precipitation. How is the degradation of precipitation defined?**

Thank you for your comment. The precipitation data depicted in Figure 2b were obtained from the ERA5 reanalysis dataset. According to the definition provided by the China Meteorological Administration, a rainstorm is typified by substantial precipitation, usually falling within the range of 50 to 100 mm per day. On July 28, the daily precipitation amounted to 90 mm, indicating that it reached the threshold for a rainstorm event. Therefore, the heavy precipitation significantly influenced the atmospheric environment during the early stage of the LD period.

Regarding the exclusion of precipitation data from the analysis in Section 3.1 of the study, we chose to exclude data collected during precipitation events to eliminate the potential impact of precipitation on the observed variables. This approach allowed for

a more precise comparison between different lockdown periods.

In response to your question about the definition of precipitation degradation, we acknowledge that precipitation can have complex and varied effects on atmospheric conditions. These effects may include changes in air quality, alterations in aerosol composition, shifts in atmospheric chemistry, scavenging of atmospheric pollutants, and interactions with atmospheric aerosols.

**3. In Section 3.5, the authors propose a method to roughly estimate the local and non-local proportions for each type of BCc particles in Yangzhou only during the ALD period. And the distribution of BCc particles during LD is unclear and there is no comparison before and after. The manuscript focuses on the COVID-19 lockdown, how do the authors provide valuable insights for future air pollution control measures? As the authors pointed out, "Since there was a heavy precipitation on July 28th (the day before lockdown) which removed most atmospheric pollutants, the air pollutants might be influenced mostly by regional transport as local emissions were significantly cut down during the LD period". Therefore, is it necessary to analyze the distribution of BCc particles during LD period?**

Thank you for your thoughtful comment. We agree with your suggestion, and upon consideration, we find that estimating the local and non-local proportions for each type of BCc particles in Yangzhou is indeed ambitious and may introduce unnecessary complexity. Therefore, we have decided to delete Section 3.5 from the manuscript.

**Minor Comments:**

**1. A detailed description of the shading in Figure 1 should be given.**

We appreciate the suggestion. We have enhanced the annotation of Figure 1 with additional details, as follows:

Line 193~196: "***Figure 1.*** *Geographical overview of the Yangtze River Delta (YRD)*

*Region in China, depicting the major cities within the YRD and the sampling site located in Yangzhou. The color gradient from green to white indicates varying altitudes across the region (Maps were generated by using Arcgis Pro).*"

**2. (Line 177-181) NO$_2$ (NRTI/L3 NO2) obtained from the TROPOspheric Monitoring Instrument (TROPOMI) with a spatial resolution of 3.5×7 km$^2$; Modern-Era Retrospective Analysis for Research and Applications, version 2 (MERRA-2 SO2SMASS) with a spatial resolution of 69×55 km$^2$. My question is, how do the authors think about the impacts of resolution difference between two datasets on the results?**

Thanks for the comment. We recognize the disparities between the spatial resolutions and units of the bands in TROPOMI and MERRA-2, which complicates the comparison of remote sensing results from these two different sensors. Consequently, we have transitioned to using the Copernicus Atmosphere Monitoring Service (CAMS) Global Near-Real-Time dataset for analyzing the distribution of NO$_2$, SO$_2$, and PM$_{2.5}$, ensuring consistency in resolution. Section 2.2.1 has been updated accordingly.

**3. (Line 228) "Dramatically", the authors should objectively describe and carefully consider the modifiers.**

Thanks for the comment. The word "dramatically" has been replaced with "significantly" as requested in Line 260.

**4. (Line 244-246) "Surface O$_3$ concentration showed an increase of 12.6 μg m$^{-3}$ (19%) during the LD period compared to the BLD period, which may attribute to the reduction of fresh NO emissions that alleviates O$_3$ titration". In fact, the O$_3$ concentration showed sustained higher values during the ALD period compare with those in BLD and LD periods (Figure 3). What is the reason?**

Thank you for the comment. We have carefully considered the factors contributing to the higher $O_3$ concentration during the ALD period. Firstly, the significantly lower duration of precipitation and rainfall during the ALD (54 hours, 40 mm) compared to the BLD (377 hours, 302 mm) and LD (202 hours, 228 mm) periods suggests a reduced removal of pollutants, including ozone, from the atmosphere through wet deposition during the ALD period. Additionally, the lower frequency of precipitation during the ALD period implies increased opportunities for atmospheric photochemical reactions, potentially leading to heightened ozone production compared to the BLD and LD periods. Furthermore, the lower average relative humidity (RH) during the ALD period (77%) compared to the BLD (84%) and LD (86%) periods contributed to decreased net ozone production, as supported by previous studies (Kavassalis and Murphy, 2017; Li et al., 2021). In conclusion, these three factors collectively contribute to the observed higher $O_3$ concentration during the ALD period.

**5. (Line 246-248) "However, the average concentrations of $PM_{2.5}$ (19.9 vs. 21.2 µg m$^{-3}$) and $SO_2$ (9.4 vs. 9.5 µg m$^{-3}$) were very close between BLD and LD stages (Figure 3)." According to this logic, TVOC was also very close between BLD and LD stages (Figure 3). Why was it not mentioned?**

Thank you for bringing this to our attention. We have incorporated the TVOC concentration into the text as requested. The revised section now reads as follows:

Line 281~283: *"However, the average concentrations of $PM_{2.5}$ (19.9 vs. 21.2 µg m$^{-3}$), $SO_2$ (9.4 vs. 9.5 µg m$^{-3}$), CO (0.61 vs. 0.64 mg m$^{-3}$) and TVOC (58 vs. 55 µg m$^{-3}$) were very close between BLD and LD stages (Figure 3)."*

**6. (Line 254) There is a discrepancy between the segmentation of the chart and the textual expression. Additionally, are all observation elements conducted at a rooftop laboratory 20 m above ground? If yes, is it appropriate to use "surface"?**

According to the "Automated Methods for Ambient Air Quality Monitoring Standard of China (HJ 664-2013)," the height of the air automatic monitoring sampling port from the ground should be within the range of 3~20 meters. Our sampling site, located in a rooftop laboratory at the fourth floor, falls within this standard requirement, with a height between 18~20 meters. Observing atmospheric pollutants at a height of 20 meters above the ground is generally considered representative of near-surface concentrations. While it may not be directly at ground level, it still provides valuable insights into pollutant levels close to the surface, where people and ecosystems are most affected.

**7. (Line 270-273) "Such results highlight the short-term, limited-scale, and human-induced reduction in air pollutants as a result of the lockdown measures in Yangzhou, and demonstrate the effectiveness of regional stringent emission control in reducing local atmospheric pollutant concentrations". From the analysis of the results, it is obvious that the result cannot be obtained. Furthermore, this conclusion is completely opposite to that in Section "Abstract" (Line 46-50).**

We apologize for the confusion caused by the original manuscript. The conclusions presented in the "Abstract" (Lines 46-50) pertain specifically to $PM_{2.5}$, suggesting that short-term, stringent local emission controls may not effectively reduce PM pollution. However, the conclusions in section 3.1 (Lines 270-273) were found lack precision and have been removed. In order to provide clarity, the text has been revised accordingly.

Line 44~47: "*Our research highlights that short-term, strict local emission controls may not effectively reduce PM pollution due to the complex generation and transmission characteristics of BCc and the non-linear responses of $PM_{2.5}$ to its precursors.*"

**8. (Line275-280) Calculations were only conducted for the regions with $PM_{2.5} > 10\ \mu g\ m^{-3}$, $NO_2 > 0.2$ Dobson units (DU), and $SO_2 > 0.2$ DU in the BLD period. What is the reason for defining these thresholds? Please provide an explanation.**

Thanks for the comment. In response, we utilized the formula (LD – BLD)/BLD) to calculate fractional changes. The initial use of thresholds in our original manuscript aimed to mitigate abnormal values of fraction change, particularly caused by pollutants with low levels during BLD (especially noticeable with TROPOMI $NO_2$ due to its higher resolution and presence of low-value pixels). Similar thresholds were also employed by Le et al. in their study (Le et al., 2020). However, in the revised manuscript, we transitioned to using the Copernicus Atmosphere Monitoring Service (CAMS) Global Near-Real-Time dataset to analyze the distribution of $NO_2$, $SO_2$, and $PM_{2.5}$. Throughout the calculation of fraction changes, we didn't encounter such issues, so we decided to remove these thresholds.

**9. (Line 297) Figure 5, please add a horizontal axis identifier.**

We are grateful for the suggestion. We have supplemented the labels for the x-axis in Figure 5.

**10. (Line 354) As shown in Figure 1, it can be seen that significant precipitation occurs during the LD period. The authors' expression here is "the day before lockdown", please unify it.**

Thank you for bringing this to our attention. We have rechecked the monitoring and meteorological data, and indeed, heavy precipitation occurred on the evening of July $27^{th}$ and early morning of July $28^{th}$, the eve of the lockdown. The text has been updated accordingly.

Line 346~348: "*During the transition of BLD to LD, heavy precipitation occurred from*

*the evening of July 27[th] to early morning of July 28[th] (the eve of lockdown), resulting in the removal of a majority of the pollutants (PM$_{2.5}$: 4 μg m$^{-3}$, O$_3$: 35 μg m$^{-3}$, NO$_x$: 8 μg m$^{-3}$)."*

**11. (Line 462, 477) RH is a key element which is responsible for the formation of BCOC and BC-SNA particles. Is there a simple positive or negative correlation between them?**

Thank you for the comment. **Figure R1** illustrates the variation in number fractions of different types of BCc particles under varying RH conditions during different lockdown periods. The data show that the majority of BC$_{aged}$ types exhibit a strong correlation with RH, while the number fractions of BC$_{fresh}$ types decline with increasing RH, indicating that higher RH environments are conducive to BC aging (Zhang et al., 2021). Specifically, sulfur-containing BCc particles (BCOC-S and BC-S) exhibit a decrease in fraction with increasing RH, whereas nitrogen-containing BCc particles (BCOC-N and BC-N) demonstrate a gradual increase in fraction with higher RH levels, eventually dominating BCc particles at elevated RH levels. However, it's important to note that the variation in number fractions of different BCc types during the BLD period significantly differs from that during the LD and ALD periods. This discrepancy may be attributed to frequent precipitation events, during which sulfate is likely removed less efficiently particularly in warmer seasons (Isokääntä et al., 2022). Consequently, the relationship between RH and the formation of BCOC and BC-SNA particles is complex and influenced by various factors. As such, it cannot be simplified to a simple positive or negative correlation.

[Figure]

**Figure R1.** Variations of number fractions of BCc particle types with relative humidity (RH) during (a) the BLD period, (b) the LD period, and (c) the ALD period.

**12. (Line 474-477) "As shown in Figure 9, BCc particles with ~400 nm Dva exhibited significant diurnal fluctuations in the OC/Cn and SNA/Cn ratios, during the LD period. Moreover, there was a noticeable increase in the proportion of BC-SNA particles during nighttime when RH was relatively high". Compared with the LD period, BCc particles exhibited more significant diurnal fluctuations in the OC/Cn and SNA/Cn ratios during the ALD period. What is the reason?**

Thank you for your insightful comment. The diurnal fluctuations observed in the OC/Cn and SNA/Cn ratios of BCc particles during the ALD period can be attributed to several factors:

a. **Changes in atmospheric conditions:** Variations in temperature, humidity, and solar radiation during the ALD period could influence the formation and transformation of OC and SNA, contributing to their diurnal fluctuations.

b. **Atmospheric photochemical processes:** Enhanced solar radiation during the ALD period could accelerate photochemical reactions involving OC and SNA, leading to their rapid formation or removal throughout the day.

c. **Changes in pollutant transport and dispersion:** Differences in atmospheric stability and boundary layer dynamics during the ALD period may result in variations in the transport and dispersion of pollutants, affecting the observed diurnal fluctuations in BCc particle composition.

**13. (Line 490-491) Please confirm the relationship between "BCc particles" and "RH" again to prevent conflicting results.**

Thank you for your comment. The diurnal variations of the $OC/C_n$ and $SNA/C_n$ ratios (Figure 9) reveal that higher relative humidity (RH) during the night facilitates the formation of a thicker coating on the surfaces of BC cores. Additionally, there is a higher fraction of $BC_{age}$ during the LD period with higher RH compared to the BLD and ALD periods (Figure 6). Furthermore, the majority of $BC_{aged}$ types exhibit a strong correlation with RH throughout the entire observation period (Figure R1). These findings demonstrate that higher RH environments promote the aging process of BCc particles.

---

## Author Response (AR2)

Responses to editor's comments

We thank the editor for the detailed, helpful, and overall supportive comments. We have revised the manuscript to account for each comment. Responses to the individual comments are provided below. Reviewer comments are in **bold**. Author responses are in plain text. Modifications to the manuscript are in *italics*. Line numbers in the response correspond to those in the revised manuscript text file.

**1. There is no clear evidence to show that the reduction of $NO_x$ was purely from the lockdown or the shift of meteorological conditions. The external information should be given to aid the reduction of human activity during the lockdown period.**

Thank you for your comment. We believe that the reduction in $NO_x$ during the lockdown was primarily due to decreased human activity rather than meteorological changes. According to the 2021 statistics from the Yangzhou Municipal Government, industrial energy consumption and transportation activity decreased by 25% and 46%, respectively, during the lockdown period compared to the same timeframe in 2020 (www.yangzhou.gov.cn). This significant reduction in key $NO_x$ emission sources strongly suggests that fewer industrial and transportation activities were the main contributors to the observed drop in $NO_x$ levels. Additionally, we have verified with the government website (www.yangzhou.gov.cn) that the lockdown in Yangzhou commenced on July 29th, 2021.

Moreover, despite meteorological conditions that typically lead to higher pollutant concentrations, such as wind speeds during the lockdown being 24% and 30% lower than in the before-lockdown (BLD) and after-lockdown (ALD) periods, and total precipitation before the lockdown being 2.6 times greater with 1.7 times more precipitation hours than lockdown, $NO_x$ levels still dropped by 39% during the lockdown. This substantial reduction in $NO_x$, despite meteorological conditions that would normally favor its accumulation, underscores that the strict lockdown measures,

which sharply curtailed human activities, were the primary factors driving the decrease in $NO_x$ concentrations.

We have updated the text, and the revised version is as follows:

Line 121~128: "*In response, stringent public health measures were imposed from July 29th to September 10th, including the closure of public transport, and suspension of non-essential industrial plants, restaurants, shopping malls, and entertainment clubs. People were also mandated to quarantine at home. Consequently, Yangzhou experienced a significant decline in transportation and industrial energy consumption, dropping by nearly 46% and 25%, respectively, compared to the same period in 2020 (www.yangzhou.gov.cn), implying a substantial reduction in human activity and primary emissions.*"

Line 147~150: "*In this study, the measurement period was divided into three phases: the before-lockdown period (BLD: 30 June to 28 July 2021), the lockdown period (LD: 29 July to 9 September 2021), and the after-lockdown period (ALD: 10 September to 7 October 2021) (**Figure 2**)*"

**2. For the question about whether the precipitation may have influenced the data, you should use the measured precipitation around your measurement site, rather than using the reanalysis results.**

Thank you for your comment. Following your suggestion, we replaced the reanalysis precipitation data with data obtained from a local weather station near the sampling site. This update ensures that the precipitation data used for analysis accurately reflects the actual conditions experienced at the site.

Additionally, to mitigate the potential impact of heavy precipitation on the data, we excluded measurements taken during a significant rainfall event prior to the lockdown period from the analysis. This adjustment helps prevent distortion of the results due to extreme weather conditions and ensures that our analysis is based on relative consistent meteorological data. We hope these changes provide clarity regarding the influence of precipitation on the data and meet your expectations.

We have updated the text, and the revised version is as follows:

Line 173~174: "*Precipitation (PCP) data was obtained from the Yangzhou Meteorological Bureau.*"

Line 232~235: "*Notably, significantly reductions in $PM_{2.5}$, $NO_x$, and $SO_2$ were observed at the end of BLD due to a high precipitation event, with a peak hourly precipitation reaching 37 mm, and the data collected during this event were excluded from the analysis.*"

**3. Following the question above, the analysis should clearly mark the above period with significant precipitations. This may affect the conclusion, as the reduction of primary pollutants may also result from the precipitation wet removal rather than the lockdown. The results may need to be reanalyzed and the discussion should be revisited.**

Thanks for the comment. we have clearly marked the periods of significant precipitation in our data (from July 25[th] to July 28[th]) to distinguish the event from the overall temporal trends. Furthermore, we have reanalyzed the results excluding the marked period of heavy precipitation and update the relative figures. The discussion section of our paper was also be revisited to incorporate these changing, ensuring that our conclusions are robust.

We have updated the text, and the revised version is as follows:

Line 349~351: "*During the transition from BLD to LD, heavy and continuous precipitation occurred from July 25[th] to July 28[th] (the eve of lockdown), resulting in the removal of a majority of the pollutants ($PM_{2.5}$: 4 $\mu g\ m^{-3}$, $O_3$: 35 $\mu g\ m^{-3}$, $NO_x$: 8 $\mu g\ m^{-3}$).*"

Line 365~367: "*Despite the abrupt reductions of NOx (-39%) due to the city lockdown, it is important to note that the concentration of PM2.5 only slightly decreased during LD (-1%), highlighting the non-linear relationship between primary emissions and PM2.5 levels.*"

Line 369~371: "*During ALD (PM2.5: 26.7 $\mu g\ m^{-3}$, NOx: 27.9 $\mu g\ m^{-3}$, TVOC: 76.0 $\mu g$*

m-3), the number fraction of BC-fresh particles rose from 28% (LD) to 31% (ALD), while the fraction of VE particles also increased from 3% (LD) to 12% (ALD) (Figure 7a)."

Line 247~265: "During LD, strict measures resulted in notably lower surface concentrations of $PM_{2.5}$ (20.3 $\mu g\ m^{-3}$), $NO_x$ (16.8 $\mu g\ m^{-3}$) and TVOC (55.9 $\mu g\ m^{-3}$) compared to BLD and ALD. Conversely, the surface $O_3$ concentration showed an increase of 18.4 $\mu g\ m^{-3}$ (28%) during LD relative to BLD. The reduction of fresh NO emission alleviates $O_3$ titration (Steinfeld, 1998) could be an explanation. Analysis from **Figure S3** indicates that the $O_3$ level is higher than those of neighboring cities in the YRD, suggesting higher local atmospheric oxidation capacity during LD. However, the average concentrations of $PM_{2.5}$ (20.6 vs. 20.3 $\mu g\ m^{-3}$), $SO_2$ (9.1 vs. 9.2 $\mu g\ m^{-3}$) and CO (0.61 vs. 0.62 $mg\ m^{-3}$) were comparable during both BLD and LD (**Figure 3**). After LD, social activities gradually resumed in Yangzhou City, leading to an apparent increase in all observed pollutants during the ALD period. For instance, there were relative increases of 66% for $NO_x$, 19% for $SO_2$, 36% for TVOC, 14% for $O_3$, 32% for $PM_{2.5}$ and 16% for CO from LD to ALD, respectively (**Figure 3**). Given that both BC and CO are byproducts of incomplete combustion of carbon-containing fuels (Wang et al., 2015), and the high correlation between BC and CO (Zhou et al., 2009), it is plausible to infer that the primary emission source of BC during LD differed from those during ALD. This change likely reflects the shift in combustion practices and fuel usage patterns as economic activities restarted during ALD."

Line 349~351: "During the transition from BLD to LD, heavy and continuous precipitation occurred from July 25$^{th}$ to July 28$^{th}$ (the eve of lockdown), resulting in the removal of a majority of the pollutants ($PM_{2.5}$: 4 $\mu g\ m^{-3}$, $O_3$: 35 $\mu g\ m^{-3}$, $NO_x$: 8 $\mu g\ m^{-3}$)."

Line 518~521: "However, the decline in NOx (-39%) and TVOC (-14%) levels might on the other hand result in increased O3 (28%), leading to a rise in BC-aged particles and a slight elevation in PM2.5 levels during the lockdown."

[Figure]

**Figure 2.** Temporal variations of **(a)** wind direction (WD) and wind speed (WS), **(b)** precipitation (PCP), **(c)** temperature (T) and relative humidity (RH), **(d)** concentrations of NO$_x$ and SO$_2$, and **(e)** mass loading of PM$_{2.5}$. The grey, blue, and red arrow ranges denote the periods before lockdown (BLD), during lockdown (LD), and after lockdown (ALD).

[Figure]

**Figure 3.** Ground-based observations of PM$_{2.5}$, NO$_x$, SO$_2$, O$_3$, CO, and TVOC concentrations in Yangzhou. The figure compares the averages during the BLD (grey), LD (blue), and ALD (red) periods. Error bars indicate SDs over different lockdown periods.

**References:**

Steinfeld, J.I., 1998. Atmospheric Chemistry and Physics: From Air Pollution to

Climate Change. Environment: Science and Policy for Sustainable Development 40, 26–26. https://doi.org/10.1080/00139157.1999.10544295

Wang, Q., Liu, S., Zhou, Y., Cao, J., Han, Y., Ni, H., Zhang, N., Huang, R., 2015. Characteristics of Black Carbon Aerosol during the Chinese Lunar Year and Weekdays in Xi'an, China. Atmosphere 6, 195–208. https://doi.org/10.3390/atmos6020195

Zhou, X., Gao, J., Wang, T., Wu, W., Wang, W., 2009. Measurement of black carbon aerosols near two Chinese megacities and the implications for improving emission inventories. Atmospheric Environment 43, 3918–3924. https://doi.org/10.1016/j.atmosenv.2009.04.062

---

## Author Response (AR3)

Public justification (visible to the public if the article is accepted and published):

Dear authors,

Please address the issues reviewer 2 raised, which I have also copied below:

"Although the authors addressed most of the concerns raised by the reviewers, I was not convinced that they fully revised the manuscript. I suggest the authors to update the line-by-line responses to the review comments so that 1) reviewers can understand how the authors tried addressing the concerns, and 2) how the authors revised the manuscript for addressing the individual points. I am unable to suggest acceptance of the manuscript in its current form.

For instance, the response to the comment on Figure 3 by the reviewer #1 is not satisfactory. The authors only tried to change some expressions, and did not seem to put efforts on checking the data carefully for addressing the concern.

the comment by reviewer #1 on line 492 was not addressed in the revised manuscript.

It is not clear how the authors addressed the comment #2 by reviewer #3 (comment about precipitation).

I was unable to identify how the authors revised the manuscript for addressing the comment #12 (comment about line 474-477) by reviewer #3. I might have missed the update. At least, it is not clarified in the line-by-line response to the review comments."

Kind regards,

Dantong Liu

Responses to editor's comments

We thank the editor for the detailed, helpful, and overall supportive comments. We have revised the manuscript to account for each comment. Responses to the individual comments are provided below. Reviewer comments are in **bold**. Author responses are in plain text. Modifications to the manuscript are in *italics*. Line numbers in the response correspond to those in the revised manuscript text file.

**1. (Figure 3) Do the authors have BC and CO concentration data? Both these chemical species are emitted from incomplete combustion. Their emission ratios depend on types of sources. If the authors could provide these data, it may help supporting the conclusion that the major emission source of BC during the LD period was different.**

Thank you for your insightful comment. The observed changes in CO concentrations and BCc classifications provide strong evidence that the major emission sources of BC during the ALD period were different from those during the LD period. The increase in vehicle emissions and changes in fuel usage patterns as economic activities resumed are key factors contributing to these changes.

We have updated the text, and the revised version is as follows:

Line 369-376: *"During ALD (PM$_{2.5}$: 26.7 $\mu g\ m^{-3}$, NO$_x$: 27.9 $\mu g\ m^{-3}$, TVOC: 76.0 $\mu g\ m^{-3}$), the number fraction of BC-fresh particles rose from 28% (LD) to 31% (ALD), while the fraction of VE particles also increased from 3% (LD) to 12% (ALD) (Figure 7a), coinciding with a 16% rise in CO concentration (Figure 3). Both BCc and CO are by-products of the incomplete combustion of carbon-based fuels and are often correlated in urban areas (Han et al., 2009). The increased CO levels, which align with the resurgence of vehicle emissions, suggest a shift in fuel usage patterns and the contribution of BCc emission sources as economic activities resumed (Wang et al.,*

*2015, Zhou et al., 2009)."*

**2. (Line 492) Could condensation also contribute to the process? Or, do the authors have a strong support to demonstrate that the process was dominantly occurring by aqueous or heterogeneous reactions?**

Thank you for the thoughtful comment. We have updated the text, and the revised version is as follows:

Line 478~490 : *"According to Surdu et al.(2023), condensation involves the direct deposition of gas-phase molecules onto the surface of particles, driven by the difference between the condensable gases concentration ($C_g$) and its equilibrium particle-phase concentration ($C_{eq}$), which is negatively affected by RH. In our study, the average RH was relatively high during all three periods (>75%), but the condensable vapor concentration decreased during the lockdown period due to strict lockdown measures, making the difference between $C_g$ and $C_{eq}$ smaller during LD compared to the other two periods. Additionally, we observed a larger mode peak (600 nm, Dva) and higher Daged/Dfresh ratios (1.11) compared to BLD (510 nm, 1.03) and ALD (500 nm, 1.02) (Figure 6). Therefore, we conclude that condensation was likely inhibited during the LD period. Instead, the conditions likely favored aqueous-phase and heterogeneous reactions, which played a more important role in the evident growth of BCc particles, converting partially coated particles into fully thickly coated BCc during the LD period."*

**3. During the observation period, precipitation occurred intermittently (Figure 2b), and the author only mentioned "the data collected during the precipitation were excluded from the analysis" in Section 3.1. This method is obviously not enough to eliminate the impact of precipitation. Additionally, the maximum daily precipitation at the observation site does not exceed 10 mm (Figure 2b), however, the author has repeatedly mentioned heavy precipitation. How is the degradation of precipitation defined?**

Following the latest editor's suggestion, we replaced the reanalysis precipitation data with data obtained from a local weather station near the sampling site, which can reflect the actual conditions experienced at the site. According to the definition provided by the China Meteorological Administration, a rainstorm is typified by substantial precipitation, usually falling within the range of 50 to 100 mm per day. On July 28, the daily precipitation amounted to 150 mm, indicating that it reached the threshold for a rainstorm event. So, we chose to exclude data collected during precipitation events to eliminate the potential impact of precipitation on the observed variables. This approach allowed for a more precise comparison between different lockdown periods. This adjustment helps prevent distortion of the results due to extreme weather conditions and ensures that our analysis is based on relative consistent meteorological data.

We have updated the text, and the revised version is as follows:

Line 173~174: "*Precipitation (PCP) data was obtained from the Yangzhou Meteorological Bureau.*"

Line 232~235: "*Notably, significantly reductions in $PM_{2.5}$, $NO_x$, and $SO_2$ were observed at the end of BLD due to a high precipitation event, with a peak hourly precipitation reaching 37 mm, and the data collected during this event were excluded from the analysis.*"

**4. (Line 474-477) "As shown in Figure 9, BCc particles with ~400 nm Dva exhibited significant diurnal fluctuations in the OC/Cn and SNA/Cn ratios, during the LD period. Moreover, there was a noticeable increase in the proportion of BC-SNA particles during nighttime when RH was relatively high". Compared with the LD period, BCc particles exhibited more significant diurnal fluctuations in the OC/Cn and SNA/Cn ratios during the ALD period. What is the reason?**

Thanks for the comment. We have added the analysis about the more significant diurnal fluctuations in the OC/Cn and SNA/Cn ratios during the ALD period , as outlined below:

Line 470~476 : *"The more significant diurnal fluctuations in the OC/Cn and SNA/Cn ratios of BCc particles during the ALD period, compared to the LD period, can be attributed to increased primary emissions from resumed society activities, more complex atmospheric chemistry involving reactive gases, and the reinstatement of typical diurnal emission patterns, with higher nighttime RH further enhancing secondary aerosol formation."*

**References:**

Han, S., Kondo, Y., Oshima, N., Takegawa, N., Miyazaki, Y., Hu, M., Lin, P., Deng, Z., Zhao, Y., Sugimoto, N., Wu, Y., 2009. Temporal variations of elemental carbon in Beijing. Journal of Geophysical Research: Atmospheres 114. https://doi.org/10.1029/2009JD012027

Surdu, M., Lamkaddam, H., Wang, D.S., Bell, D.M., Xiao, M., Lee, C.P., Li, D., Caudillo, L., Marie, G., Scholz, W., Wang, M., Lopez, B., Piedehierro, A.A., Ataei, F., Baalbaki, R., Bertozzi, B., Bogert, P., Brasseur, Z., Dada, L., Duplissy, J., Finkenzeller, H., He, X.-C., Höhler, K., Korhonen, K., Krechmer, J.E., Lehtipalo, K., Mahfouz, N.G.A., Manninen, H.E., Marten, R., Massabò, D., Mauldin, R., Petäjä, T., Pfeifer, J., Philippov, M., Rörup, B., Simon, M., Shen, J., Umo, N.S., Vogel, F., Weber, S.K., Zauner-Wieczorek, M., Volkamer, R., Saathoff, H., Möhler, O., Kirkby, J., Worsnop, D.R., Kulmala, M., Stratmann, F., Hansel, A., Curtius, J., Welti, A., Riva, M., Donahue, N.M., Baltensperger, U., El Haddad, I., 2023. Molecular Understanding of the Enhancement in Organic Aerosol Mass at High Relative Humidity. Environ Sci Technol 57, 2297–2309. https://doi.org/10.1021/acs.est.2c04587

Wang, Q., Liu, S., Zhou, Y., Cao, J., Han, Y., Ni, H., Zhang, N., Huang, R., 2015. Characteristics of Black Carbon Aerosol during the Chinese Lunar Year and Weekdays in Xi'an, China. Atmosphere 6, 195–208. https://doi.org/10.3390/atmos6020195

Zhou, X., Gao, J., Wang, T., Wu, W., Wang, W., 2009. Measurement of black carbon aerosols near two Chinese megacities and the implications for improving emission inventories. Atmospheric Environment 43, 3918–3924. https://doi.org/10.1016/j.atmosenv.2009.04.062